



# Air-sea gas exchange at hurricane wind speeds

Kerstin E. Krall[1] and Bernd Jähne[1,2]

[1]Institute of Environmental Physics, Heidelberg University, Im Neuenheimer Feld 229, 69120 Heidelberg, Germany
[2]Heidelberg Collaboratory for Image Processing, Heidelberg University, Berliner Straße 43, 69120 Heidelberg, Germany

**Correspondence:** K. E. Krall (kerstin.krall@iup.uni-heidelberg.de)

**Abstract.** Gas transfer velocities were measured in two high-speed wind-wave tanks (Kyoto University and the SUSTAIN facility, RSMAS, University of Miami) using fresh water, simulated seawater and seawater for wind speeds between 7 and $80\,\mathrm{m\,s^{-1}}$. Using a mass balance technique, transfer velocities of a total of 12 trace gases were measured, with dimensionless solubilities ranging from 0.005 to 150 and Schmidt numbers between 149 and 1360. This choice of tracers allowed to separate

gas transfer across the free interface from gas transfer at closed bubble surfaces. The major effect found was a very steep increase of the gas transfer across the free water surface at wind speeds beyond $33\,\mathrm{m\,s^{-1}}$, which is the same for fresh water, simulated seawater and seawater. This steep increase might start at a lower wind speed in the open ocean as compared to the short-fetch wind-wave tanks. Bubble-induced gas transfer plays no significant role for all tracers in fresh water and for tracers with moderate solubility such as carbon dioxide and DMS in seawater, while for low solubility tracers bubble-induced gas

transfer in seawater was found to be about 1.7 times larger than the transfer at the free water surface at the highest wind speed of $80\,\mathrm{m\,s^{-1}}$.

## 1 Introduction

The transfer of trace gases across the air-sea interface has been an active field of research for almost 40 years (Jähne, 2019).

The transfer is characterized by the gas transfer velocity, which depends on environmental forcing such as the wind speed, the amount and strength of wave breaking, the presence of surface active material, number and size of bubbles and spray created by breaking waves (Wanninkhof et al., 2009; Jähne, 2019).

Measuring the gas transfer velocity under hurricane conditions in the field is extremely challenging. Using unmanned floats, McNeil and D'Asaro (2007) managed to measure three gas transfer velocities at wind speeds higher than $25\,\mathrm{m\,s^{-1}}$ during

Hurricane Frances in 2004. Due to the difficulties of measuring in the field, wind-wave tanks capable of producing hurricane force winds are a viable and safe alternative, as they allow for a controlled study of the air-sea interaction mechanisms up to the highest wind speeds.

Until now, only two gas transfer studies have been performed in hurricane conditions in the Kyoto high-speed wind-wave tank with 1,4-difluorobenzene and hexafluorebenzene (Krall and Jähne, 2014) and with carbon dioxide (Iwano et al., 2013,



2014), but only in fresh water. Both studies found a strong increase in the gas transfer velocity at wind speeds higher than approximately $35\,\mathrm{m\,s^{-1}}$. Gas transfer was found to increase with more than the third power of the wind speed. However, because of the few gases used, it remains unclear, which process is the main cause of this steep increase. Possible candidates are a) bubbles, which provide an additional surface for the gas transfer, b) an increased water surface area due to the fragmentation of the water surface at highest wind speeds, or c) a strong increase in near surface turbulence due to frequent surface renewal and breakup events, or a combination of all three effects. It is also not clear whether bubble-induced gas exchange differs between fresh water and seawater.

This paper reports the results of extensive gas exchange measurements in two different wind-wave tanks with up to 12 tracers covering a wide range of solubilities using fresh water, simulated seawater and seawater.

## 2 Air-sea interactions

### 2.1 Gas Transfer

The flux density $j$ of a trace gas across the air-sea interface is governed by the difference in concentration of the gas in air and water ($c_a$ and $c_w$) and the gas transfer velocity in water $k_s$ across the water surface,

$$j = k_s \Delta c = k_s(c_w - \alpha c_a).$$

Because of the discontinuity at the air-water boundary, the solubility $\alpha$ (here, $\alpha$ is equal to the dimensionless Henry solubility $H^{cc}$, (Sander, 2015)) has to be taken into account. The gas transfer velocity $k_s$ of a sparingly soluble tracer through a free, smooth or wavy, unbroken surface can be described by

$$k_s = \frac{1}{\beta} u_* \mathrm{Sc}^{-n} \tag{1}$$

(Jähne et al., 1989) with the friction velocity $u_*$, a measure for momentum input into the water by the wind, the Schmidt number $\mathrm{Sc} = \nu/D$ of a tracer, given by the ratio of the kinematic viscosity of water $\nu$ and the tracer's diffusion coefficient in water $D$. The dimensionless parameter $\beta$ and the Schmidt number exponent $n$ depend on the boundary conditions, with $n = 2/3$ for a smooth water surface and $n = 1/2$ for a rough and wavy surface.

From Eq. 1 it is apparent, that the transfer velocities of two sparingly soluble gases A and B can be converted by Schmidt number scaling,

$$\frac{k_{s,A}}{k_{s,B}} = \left(\frac{\mathrm{Sc}_A}{\mathrm{Sc}_B}\right)^{-n}. \tag{2}$$

Commonly, a reference Schmidt number of $\mathrm{Sc} = 600$ is chosen, which corresponds to carbon dioxide at $20^oC$ in fresh water.

For gases, that have a medium to high solubility, the transfer resistance in the air side has to be taken into account. As first shown by Liss and Slater (1974) the total transfer velocity $k_t$ can then be expressed by

$$\frac{1}{k_t} = \alpha \frac{1}{k_a} + \frac{1}{k_s} \tag{3}$$





with $k_a$ being the air-side transfer velocity. For gases with a low solubility, the second term dominates and the first term in Eq. 3 can be neglected, such that for those gases $k_t = k_s$. Inverse transfer velocities can be seen as transfer resistances, such that Eq. 3 can be written as

$$R_t = \alpha R_a + R_s. \tag{4}$$

All transfer velocities used in this paper are related to a water side observer, i.;e., describe how fast a gas in transferred into or out of the water. Air-side observed gas transfer velocities differ by a factor of $\alpha$.

## 2.2 Bubble mediated gas transfer

Bubbles contribute to the gas transfer in two ways. First, they provide an additional surface through which gases can pass. Second, during their generation, by rising through the water side mass boundary layer of the water surface, and by bursting

at the water surface, they increase the near surface turbulence. Monahan and Spillane (1984) already considered whitecaps as 'low impedance vents', which 'shortcut' the water side transfer resistance. These bubble effects which intensify near surface turbulence increase the transfer velocity across the free surface and do not depend on tracer solubility.

The transfer through a closed bubble surface is different from transfer across the free water surface. First, bubbles have a limited life time, as they either burst at the water surface or, if they are small enough, completely dissolve. Second, bubbles

have a limited volume to take up or release gas. Once a bubble is filled to the equilibrium concentration $c_b^{eq}$ given by Henry's law, $c_b^{eq} = \alpha^{-1} c_w$, it is inactive for the remainder of its life time. For gases with higher solubilities and for small bubbles, this equilibrium is reached faster. And third, bubbles experience an overpressure due to hydrostatic pressure and surface tension. Therefore small bubbles can completely dissolve and the equilibrium concentration shifts to slightly higher concentrations. Because the measurements reported here are taken far from equilibrium, dissolving bubbles are not important. The transfer

velocities themselves are not affected. A detailed analysis of the time scales involved and how they depend on the bubble radius is given in Jähne et al. (1984).

Because bubbles form an additional exchange surface, the total water side transfer velocity $k_w$ can be split up into two parts (Merlivat and Memery, 1983; Goddijn-Murphy et al., 2016),

$$k_w = k_s + k_c \tag{5}$$

with transfer through the free water surface $k_s$ and through the closed surface of submerged bubbles $k_c$. It is important to note here that the bubble-induced gas transfer velocity $k_c$ does not include the bubble formation process and the bursting of bubbles when they rise through the surface again. Concerning the gas transfer velocity, these effects cannot be distinguished from other processes generating turbulence close to the water surface, because they do not depend on tracer solubility but only on the Schmidt number. Therefore bubble-induced gas exchange refers here only to the stages in the life time of a bubble with

a closed surface and therefore a limited trapped air volume.

Figure 1 shows a schematic view of the resistances for bubble mediated gas transfer $R_c = k_c^{-1}$ in relation to the air and water sided resistances for transfer through the unbroken water surface.




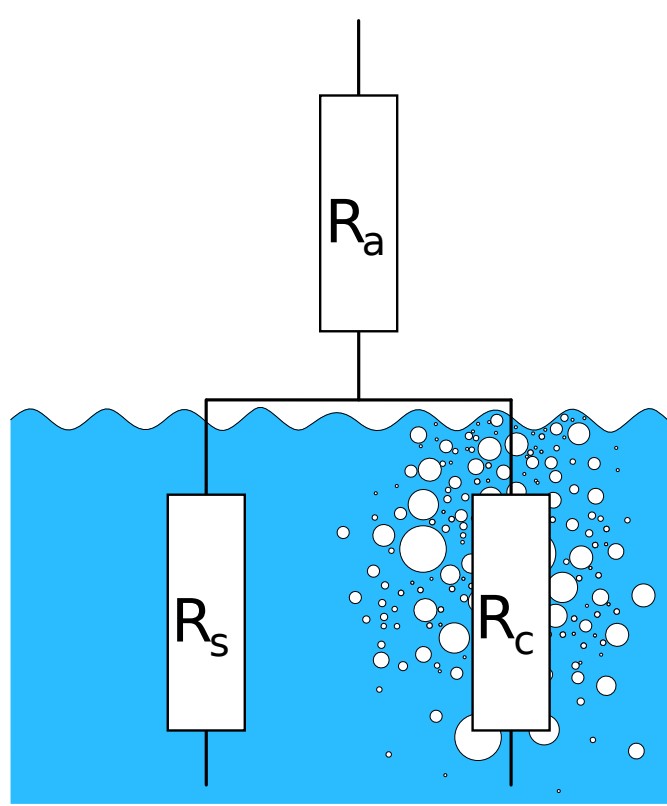

**Figure 1.** Transfer resistances: $R_a$: air side transfer resistance, $R_s$: water side resistance of the free water surface, $R_c$: transfer resistance of the closed bubble surfaces.

Maiß (1986)

Many approaches have been made to quantify the bubble mediated gas transfer $k_c$: gas transfer by single bubbles (Maiß, 1986; Patro et al., 2002), transfer in bubble clouds (Asher et al., 1996; Mischler, 2014) and breaking waves (Asher et al., 1995; Leifer and De Leeuw, 2002) as well as theoretical models based on bubble dynamics (Memery and Merlivat, 1985; Woolf and Thorpe, 1991). Bubble mediated gas transfer also depends on bubble surface conditions. It has been shown that surface active material reduces the gas transfer of single bubbles (Maiß, 1986; Patro et al., 2002), while it also decreases the bubbles rise velocity (Alves et al., 2005). During the lifetime of a bubble these surface conditions can change, as bubbles accumulate surface active material while moving through the water.

Empirical or semi-empirical parameterizations (Goddijn-Murphy et al., 2016; Woolf et al., 2007) are state of the art of calculating the bubble mediated gas transfer $k_b$ on the open ocean. Most of these parameterizations link $k_b$ to the tracer's solubility and Schmidt number (or diffusion coefficient) as well as the whitecap coverage of the water surface, which in turn depends on the sea state that is usually expressed as a function of the wind speed in 10 m height $u_{10}$ or the friction velocity $u_*$.





Physically based models (Woolf et al., 2007; Mischler, 2014) distinguish two limiting cases, one for very weakly soluble gases and one for more highly soluble ones. For very weakly soluble tracers the bubbles act as a very big reservoir. In that case the bubbles simply provide an additional surface for gas transfer that actively participates in gas transfer for the whole lifetime of a bubble. In this limit, the gas transfer is proportional to the integrated bubble surface area $A_{b,\delta r}$ per radius interval $\delta r$ normalized to the water surface area $A_s$, and the Schmidt number with the bubble Schmidt number exponent $n_b$, and does not depend on solubility,

$$k_{c,\text{low}\,\alpha} = \frac{\int A_{b,\delta r}(r) k_{b,600}(r) dr}{A_s} \left(\frac{600}{\text{Sc}}\right)^{n_b} = k_{c,600} \left(\frac{600}{\text{Sc}}\right)^{n_b}. \tag{6}$$

The transfer velocity $k_c$ is the effective bubble-induced transfer velocity related to the free water surface, while $k_b(r)$ is the real transfer velocity across the bubble surface of a given radius.

In the limit of high solubility, the bubbles constitute a very small reservoir for the trace gas, so that the higher solubility tracers will reach concentration equilibrium $c_b^{eq} = \alpha^{-1} c_w$ very fast. Then the bubble mediated gas transfer can only depend on the tracer's solubility and the total bubble volume flux $Q_{b,\delta r}$ per radius interval $\delta r$,

$$k_{c,\text{high}\,\alpha} = \frac{1}{\alpha} \frac{\int Q_{b,\delta r}(r) dr}{A_s} = \frac{1}{\alpha} \frac{Q_b}{A_s} = \frac{1}{\alpha} k_r. \tag{7}$$

The velocity $k_r$ has an intuitive meaning. It is the effective velocity (volume flux per water surface area) averaged over all bubble sizes, with which the air volume is being submerged by breaking waves and rises towards the surface again.

The transition solubility, at which the constant bubble mediated transfer velocity for low solubility, $k_{c,\text{low}\,\alpha}$ changes into the transfer velocity decreasing with increasing solubility can be computed by setting both values equal:

$$\alpha_t = \frac{k_r}{k_{c,600}} \left(\frac{\text{Sc}}{600}\right)^{n_b}. \tag{8}$$

Based on detailed measurements in a bubble-tank with multiple volatile tracers, Mischler (2014) showed that the transition between the two regimes can be well described by a simple exponential term (Fig. 2). The transfer velocity for bubble-mediated gas exchange results in

$$k_c = \frac{1}{\alpha} k_r \left[ 1 - \exp\left( -\frac{\alpha}{\alpha_t} \right) \right]. \tag{9}$$

For fresh water Mischler (2014, Table 9.7) found under the conditions shown in Fig. 2 a transition solubility $\alpha_t = 0.23$ and 0.06 for seawater.

In summary, the total gas transfer velocity $k_{tot}$ for water side controlled tracers including bubble-mediated gas transfer can be parameterized as

$$k_{tot} = k_s + k_c = k_{s,600} \left(\frac{600}{\text{Sc}}\right)^{0.5} +$$

$$+ \frac{1}{\alpha} k_r \left[ 1 - \exp\left( -\frac{\alpha k_{c,600}}{k_r} \left(\frac{600}{\text{Sc}}\right)^{0.5} \right) \right], \tag{10}$$
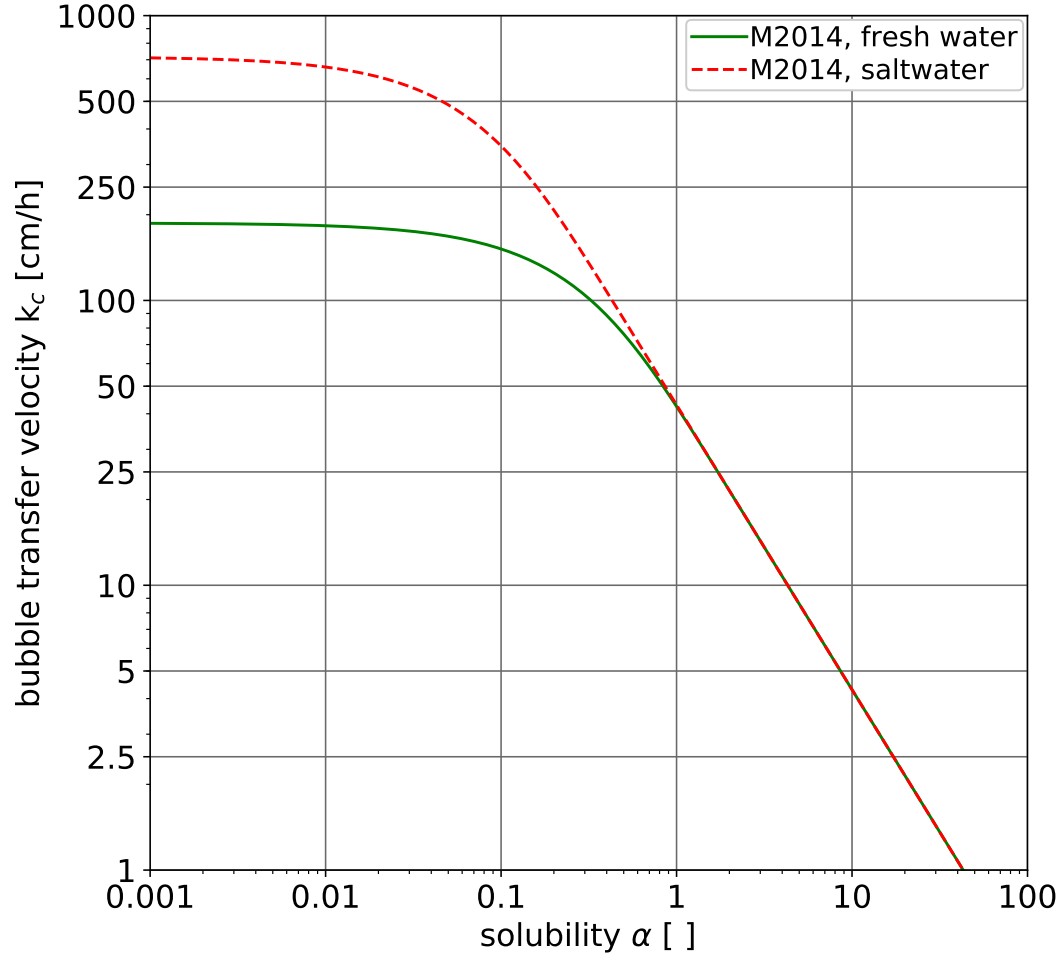

**Figure 2.** Dependency of bubble mediated gas transfer velocities for the tracers covering a wide range of solubilities as measured in a bubble tank, in which breaking waves were simulated by a water jet for fresh water and seawater (Mischler, 2014). The jet energy was 3.3 W and the bubble volume flux kept constant at about $750\,\mathrm{ml\,min^{-1}}$, corresponding to $k_r = 43\,\mathrm{cm\,h^{-1}}$.





with the three parameters $k_{\mathrm{s},600}$, $k_{\mathrm{c},600}$ and $k_r$. Because the measurements were performed with clean water, the Schmidt number exponents for both the transfer across the free water surface and the bubbles surfaces are set to 1/2. For tracers with a low solubility ($\alpha \ll \alpha_t$), the model equation (Eq. 10) can be simplified to

$$k_{tot} = (k_{\mathrm{s},600} + k_{\mathrm{c},600}) \left(\frac{600}{\mathrm{Sc}}\right)^{0.5}. \tag{11}$$

In this case it is possible to simplify the analysis, because the gas transfer rate does no longer depend on solubility and simple Schmidt number scaling can be applied. The ratio of gas transfer across bubble surfaces and the free surface is then simply given by the ratio of $k_{\mathrm{c},600}$ to $k_{\mathrm{s},600}$.

## 2.3 Spray mediated gas transfer

The processes mirroring bubbles in the water, spray droplets in the air, is less well studied, with the exception of the transfer
of water vapor and heat (Mestayer and Lefauconnier, 1988; Andreas and Emanuel, 2001; Zheng et al., 2008). Only recently, Andreas et al. (2017), evaluated the time scales governing spray-mediated gas transfer for gases other than water vapor in a similar fashion as Jähne et al. (1984) did for bubble-mediated transfer more than three decades earlier.

In contrast to bubbles, tracer solubility plays no role for spray droplets as long as the transfer process is controlled by water-sided processes. This is the case for all tracers used. Therefore spray droplets just constitute an additional exchange surface.
The question is only, whether the life time for gas exchange is longer than the life time of the droplets. If this is the case, gas exchange occurs during their whole life time. In this upper limit the gas transfer velocity $k_d$ induced by spray droplet is given in analogy to Eq. 6 by

$$k_{\mathrm{d,upper}} = \frac{\int A_{d,\delta r}(r) k_{\mathrm{d},600}(r) dr}{A_s} \left(\frac{600}{\mathrm{Sc}}\right)^{n_d} = k_{\mathrm{d},600} \left(\frac{600}{\mathrm{Sc}}\right)^{n_d}. \tag{12}$$

The transfer velocity $k_{\mathrm{d}}$ is the effective droplet-induced transfer velocity related to the free water surface, while $k_d(r)$ is the
real transfer velocity across the droplet surface of a given radius.

If the concentration inside the spray droplet equilibrates faster with the surrounding air than the spray droplet takes to fall back into the water, the spray-induced gas transfer velocity $k_d$ depends on the total volume flux $Q_d$ of spray generated (Andreas et al., 2017). This lower limit is given by

$$k_{d,\mathrm{lower}} = Q_d/A_s. \tag{13}$$

At the highest wind speeds, water is lost from wind-wave tanks because the wind tears offs the wave crests and part of the resulting spray droplets leave the facility with the air flow (Sec. 3.2). Therefore the volume lost $\dot{V}_w$ (see Sec. 3.2) is actually a lower limit for $Q_d$.

Because solubility plays no role for the tracers used in the experiment here, spray-induced gas transfer cannot be distinguished from gas transfer across the free water surface. Another effect may happen, however. In the limit of a long droplet
life time compared to the life time for gas exchange (Eq. 13), the spray droplet induced gas exchange does not depend on the





Schmidt number. According to the estimates of Andreas et al. (2017) this is the case. Then, gases with a high diffusivity will no longer show correspondingly higher transfer velocities, if gas transfer through the spray droplet surface is significant.

## 2.4 Drag coefficient limitation at very high wind speeds

At very high wind speeds, breaking waves disrupt the water surface. It has been shown that the drag coefficient $C_d = u_*^2 u_{10}^{-2}$
5 gets saturated or even decreases at wind speeds higher than around $30 - 35\,\mathrm{m\,s^{-1}}$ (Donelan, 2018; Soloviev and Lukas, 2010; Powell et al., 2003). A two phase layer forms, consisting of bubble-filled water transitioning to spray-filled air. The turbulence characteristics of this two phase layer is thought to be controlling the transfer of momentum, which leads to the saturation of the drag coefficient (Soloviev and Lukas, 2010). However, this does not mean that the friction velocity and thus the momentum input from the wind into the water also decreases, it just increases less steeply.

10 ## 3 Methods

### 3.1 The wind-wave tanks

Measurements were performed in two wind wave tanks, the High Speed Wind-Wave Tank of Kyoto University, Kyoto, Japan, in October of 2015 and the SUrge STructure Atmosphere INteraction Facility (SUSTAIN), University of Miami, Miami, USA in May and June of 2017. Table 1 gives an overview of the technical data of the facilities.

**Table 1.** Technical data of the wind-wave tanks used. All numbers are approximate. The water volume in parentheses for the Kyoto tank gives the total volume when the external tank was used in addition during the highest wind speed condition. Wind speeds and water temperature given in parentheses for the Kyoto experiments is for the seawater model.

|  | Kyoto Wind-Wave Tank | SUSTAIN |
|---|---|---|
| water volume [m³] | 8.5(13.7) | 120 |
| width [m] | 0.8 | 6 |
| total length [m] | 15.7 | 24 |
| length affected by wind [m] | 12.9 | 18 |
| typical water level [m] | 0.75 | 0.85 |
| air space height [m] | 0.85 | 1.15 |
| water surface area affected by wind [m²] | 10.3 | 108 |
| wind speeds [m s⁻¹] | 7–67 (41–67) | 14–80 |
| water temperature range [ºC] | 16.0–19.5 (12.8–15) | 25.0-27.4 |



**Water types**

Due to technical limitations, seawater could not be used in the Kyoto High Speed Wind-Wave Tank. There, one set of experiments was performed with tap water (referred to as fresh water or FW hereinafter). A second set of experiments was performed in Kyoto with a small amount of n-Butanol added (approx. 700 ml) to the tap water, which modifies the bubble spectrum to
better resemble that of seawater (Flothow, 2017). This second set of experiments will be referred to as seawater model or SWM.

In the SUSTAIN tank, filtered seawater taken from Biscayne Bay was used. This set of experiments will be abbreviated by SW.

### 3.2 Mass balances for evasion experiments

A mass balance method is used to measure the gas transfer velocities in evasion type experiments. In this approach, all gases are dissolved in the water and the water is mixed well by pumps, before the wind is turned on. When the wind is turned on, the main flux is from the water volume to the air, and thus the concentration of the tracer in the water $c_w$ decreases exponentially,

$$c_w(t) = c_w(0) \exp\left(-k \cdot \frac{A}{V_w} t\right), \tag{14}$$

with the water volume $V_w$, the water surface $A$ and the concentration at the start of the experiment $c_w(0)$. In the Kyoto High
Speed tank, the water lost due to spray was replaced with fresh water, which changes Eq. 14 to

$$c_w(t) = c_w(0) \exp\left(-\left(k \cdot \frac{A}{V_w} + \frac{\dot{V}_w}{V_w}\right) \cdot t\right), \tag{15}$$

with the water loss rate $\dot{V}_w/V_w$. Thus, knowing the water volume, water surface area and the loss rate and measuring a concentration time series allows to determine the gas transfer velocity $k$. A more thorough derivation of Eq. 15 can be found in Krall and Jähne (2014).

### 3.3 Gas concentration measurements, gas handling and tracers

In both experimental campaigns a dual membrane inlet mass spectrometer (HPR-40 MIMS, Hiden Analytical, Warrington, UK) was used to measure the tracers' concentrations in water. The water extracted from the wind-wave tank was pumped along one of the inlet membranes, where dissolved species diffuse through the membrane directly into the vacuum of the spectrometer where they are ionized and subsequently analyzed with respect to their concentrations. For some tracers, two mass to charge
ratios were monitored with the MIMS, either because there are sufficiently high concentrations of different Isotopes (e. g. Xe), or the tracer molecule is destroyed in the ionization process and forms multiple ions with a different masses (e. g. DMS, DFB, HFB).

As mentioned in the previous section, before the start of the evasion experiment, all available gases were dissolved in the water and mixed well. For dissolving gases into the water, a membrane contactor (SUSTAIN: Liqui-Cel 8x20PVC Kyoto:





**Figure 3.** Schmidt number - solubility diagram of the tracers used in this study. Also shown as a blue line is for which solubility the air side resistance is equal to the water side resistance (Eq. 4 for a rough water surface). Diamonds: fresh water, circles: seawater model, squares: seawater. The variations in the Schmidt number and solubility result from the varying water temperatures used in the different experiments, see Table 1.



Liqui-Cel 4x13, Membrana 3M, Charlotte, NC, USA) was used. In Miami, the gases were dissolved directly into the water of the wind-wave tank, while in Kyoto the gases were first dissolved into a holding tank of approx. $7\,\mathrm{m}^3$. This water was then mixed into the main water volume of the wind-wave tank using pumps before the start of an experiment. Care was taken that the tracers were mixed into the water as homogeneously as possible. To achieve this, the pumps were kept on even after gas

5   loading was finished, and the concentration was monitored. Only when the concentration was sufficiently stable the pumps were turned off and the experiment was started.

The tracers were chosen in this study to cover a wide range of solubilities and Schmidt numbers. Table 2 gives an overview of the tracers used sorted by their solubility. Due to technical and logistical reasons, not all tracers could be used in both facilities. Figure 3 shows the tracers in a Schmidt number–solubility diagram for all conditions encountered. The temperature

10   dependency of the solubility and Schmidt number is apparent. Also shown is for which solubility-Schmidt number combination the air side resistance equals the water side resistance ($\alpha R_a = R_s$, see also Eq. 4). To calculate the resistances, a rough and wavy water surface was assumed (i. e. $n = 1/2$). Below this $\alpha R_a = R_s$ line, the water side resistance dominates, therefore, the tracers are called water side controlled tracers. All tracers used in this study can be classified as water side controlled tracers, with the exception of Methylacetate (MA), which is partially air side controlled due to its relatively high solubility.

**Table 2.** Tracers used in this study. PFE: $C_2HF_5$, HFB: hexafluorobenzene, DFB: 1,4-difluorobenzene, DMS: dimethyl sulfide, MA: methyl acetate. Solubility and Schmidt number are given at $20^oC$ for fresh water. Schmidt numbers were calculated from the kinematic viscosity (Kestin et al., 1978) and the diffusion coefficient given in the respective citation. ([*] measured only in seawater. [†] measured only in fresh water. [‡] measured only in fresh water and seawater model.)

| Tracer | Solubility | Schmidt number |
|---|---|---|
| $CF_4$[‡] | $0.0045$[a] | $812$[h] |
| $SF_6$ | $0.0049$[a] | $950$[i] |
| He | $0.0092$[b] | $149$[j] |
| $Kr$[*] | $0.055$[b] | $624$[j] |
| PFE | $0.072$[c] | $1030$[h] |
| $Xe$[†] | $0.092$[b] | $789$[j] |
| $C_2H_2$[*] | $0.91$[d] | $686$[h] |
| HFB | $1.1$[e] | $1360$[h] |
| $CH_2F_2$ | $1.5$[f] | $818$[h] |
| $DFB$[‡] | $3.1$[e] | $1230$[h] |
| DMS | $11.2$[a] | $983$[h] |
| MA | $150$[g] | $856$[h] |

[a] (Warneck and Williams, 2012); [b] (Abraham and Matteoli, 1988); [c] (Reichl, 1995); [d] (Sander et al., 2011); [e] (Hiatt, 2013); [f] (Maaßen, 1995); [g] (Fenclová et al., 2014); [h] (Yaws, 2014); [i] (King and Saltzman, 1995); [j] (Jähne et al., 1987)





### 3.4 Wind speed measurements

**Kyoto experiments**

In the Kyoto tank, wind speeds were not recorded during the experiments. Takagaki et al. (2012) describes the measurements of the reference wind speed in 10 m height $u_{10}$ and of the friction velocity $u_*$ in the Kyoto Tank. Since we used the same wind

generator settings as Takagaki et al. (2012) wind speed and friction velocities were taken from there.

**SUSTAIN experiments**

A sonic anemometer (IRGASON, Campbell Scientific Inc., Logan, USA) was mounted in the test section of the SUSTAIN wind wave tank. The measured wind speeds were converted to the friction velocity $u_*$ and the wind speed $u_{10}$ using the parameterization for the Drag coefficient and the roughness length given in Powell et al. (2003). Wind speed uncertainty was

calculated from the device uncertainty as specified by the manufacturer as well as the variance of the wind speeds measured. Uncertainties found were in the order of 3 to 4%.

### 3.5 Bubble measurements

Bubble size distributions were measured in Kyoto using an optical bright field imaging technique (Mischler and Jähne, 2012; Flothow, 2017). In this method, a camera looks perpendicular to the wind direction through the water into a light source.

Bubbles entrained in the water scatter the light such that the light no longer reaches the camera sensor, and the bubble appears as a dark circle. Out of focus bubbles have a blurry edge, which is used to estimate the 3D volume the bubbles are in in the 2D images taken by the camera (depth from focus). Two bubble imaging systems consisting of a Nikon D800 digital single lens reflex camera and a purpose built LED light source each were operated during the measurements in Kyoto, one at a fetch of 3 m, the second one at 8 m fetch, approximately 30 cm below the water surface. Calibration and data evaluation is described in

detail in Flothow (2017).

## 4 Results

### 4.1 Wind speeds and friction velocities

The relationship between the water sided friction velocities $u_{*,\mathrm{w}}$ and the wind speeds in 10 m height, $u_{10}$ at which the gas transfer velocities were measured in this study is shown in Fig. 4a. A clear change in the steepness of the relationship can be

seen at a wind speed of approximately $33\,\mathrm{m\,s^{-1}}$ as indicated by the gray line. The wind speed of $33\,\mathrm{m\,s^{-1}}$ corresponds to a friction velocity of about $6\,\mathrm{cm\,s^{-1}}$. Also, the Drag coefficient $C_{\mathrm{D}}$ (Fig. 4b) has a maximum at this wind speed, before it levels off.

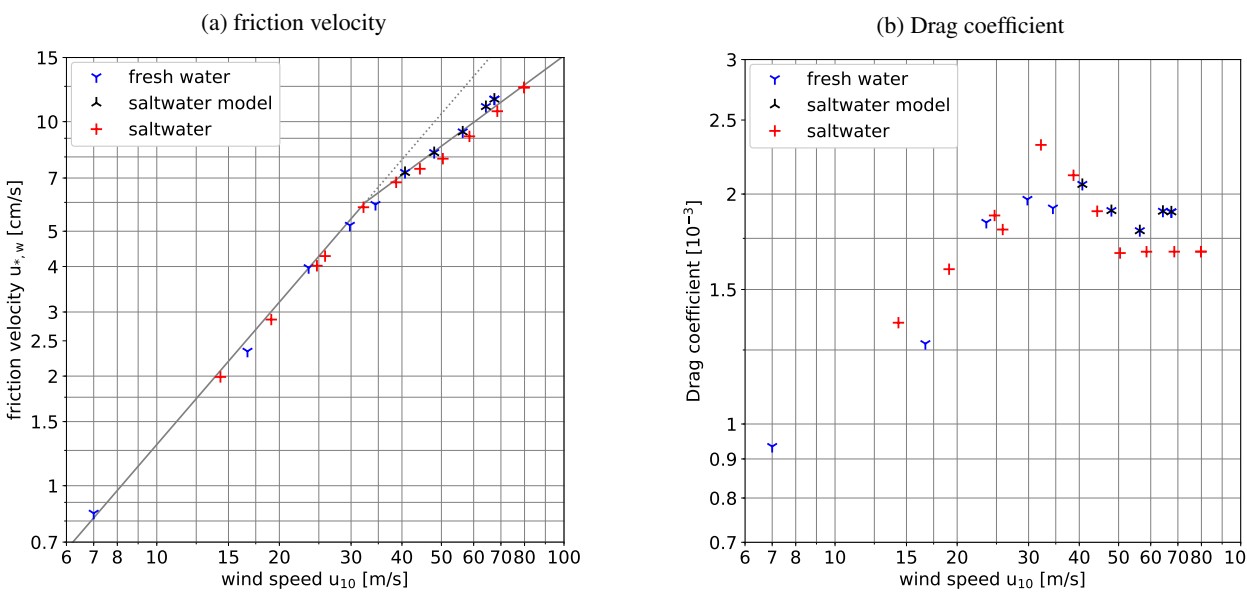

**Figure 4.** Relationship between the wind speed in 10 m height, $u_{10}$, and the water side friction velocity (a) and the Drag coefficient (b)

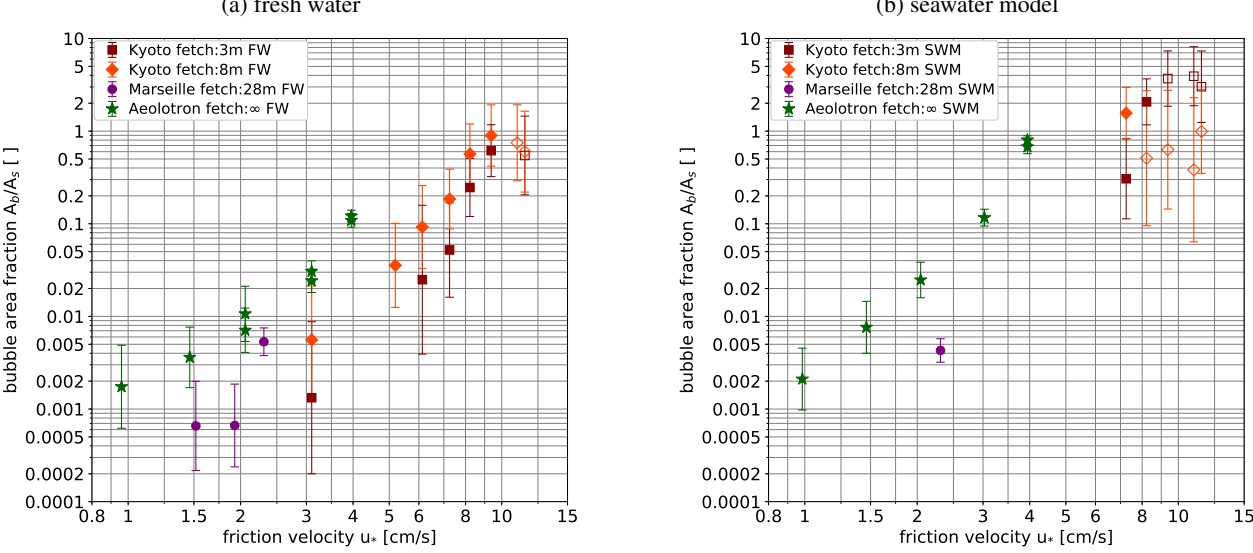

**Figure 5.** Bubble area per water surface area measured in fresh water (a) and seawater modeled by adding butanol to fresh water (b): Kyoto: 3 and 8 m fetch, camera installed approx. 30 cm below water surface at rest; Marseille Luminy: 28 m fetch, camera installed at approx. 50 cm below water surface at rest; Aeolotron Heidelberg: infinite fetch, camera installed at 10 evenly spaced positions between 0.5 and 36.5 cm below the water surface. At the highest wind speeds, the density of the bubbles is so large that the intensity of the illumination that reaches the camera reduces by up to 75 %, which leads to a systematic underestimation of the bubble surface area. Conditions likely affected by this are marked with an open symbol. Figure reproduced from data by Flothow (2017).





## 4.2 Bubble surface area

Up to now, bubble measurements were evaluated only for the Kyoto facility. From the bubble concentration measurements, the total bubble surface was computed and plotted in Fig. 5 as a dimensionless area in relation to the flat free water surface area, $A_b/A_s$. The uncertainties are quite high, because bubbles where measured only at one depth (Sec. 3.5). But still a few

important findings can be stated.

1. The bubble surface area strongly increases with the friction velocity ($\propto u_{*,\mathrm{w}}^3$ to $\propto u_{*,\mathrm{w}}^4$) in all facilities and for fresh water and seawater.

2. For fresh water the bubble surface area almost reaches the same area as the flat free surface at the highest wind speeds.

3. In the modeled seawater the bubble surface area is about on order of magnitude higher than in fresh water. At higher

friction velocities, especially with the seawater model, the bubble clouds get very dense resulting in a systematic underestimation of the bubble surface area. Therefore the true bubble surface area at the highest wind speed is very likely larger than the measured surface area of about four times the area of the flat free water surface.

4. The bubble surface area shows a clear trend to increase with fetch. Also shown in Fig. 5 are measurements performed at 28 m fetch (Marseille tank), and quasi infinite fetch in the annular wind-wave facility, the Heidelberg Aeolotron in

addition to the measurements in Kyoto at 3 and 8 m fetch. Roughly, the same bubble surface area is obtained at about half the friction velocity in the Aeolotron as compared to 3 m fetch in the Kyoto facility. With the modeled sea water the bubble surface area becomes equal to the flat free surface area at a friction velocity just above $4\,\mathrm{cm\,s^{-1}}$, while this value is reached in the Kyoto facility only at a friction velocity of about $8\,\mathrm{cm\,s^{-1}}$. This finding is important when extrapolating laboratory results to the field and will be discussed further in Sec. 4.6.

## 4.3 Measured gas transfer velocities

Figure 6 shows the measured gas transfer velocities $k_{\mathrm{meas}}$ in dependency of the water-sided friction velocity $u_{*,\mathrm{w}}$ for three different measurement conditions: (a) fresh water in Kyoto, (b) seawater model in Kyoto and (c) seawater in Miami. Gas transfer increases strongly with the wind speed for all tracers. Beyond a friction velocity of approx. $6\,\mathrm{cm\,s^{-1}}$, the increase is significantly steeper.

The tracer with the highest solubility, Methylacetate (MA), shows a transfer velocity significantly lower than for the other tracers for all wind speeds and all water types. This reduction in the gas transfer velocity confirms the existence of an additional air side resistance for this tracer, see Sec 2.1. Due to this additional air side resistance, MA will be excluded in the following discussion. From Fig. 6 it is also evident that He has a much higher transfer velocity than the other tracers for all wind speeds. This is caused by its significantly higher diffusion coefficient corresponding to a low Schmidt number, while all other tracers

vary in the Schmidt number by at most a factor of two, see Table 2.

    Figure 6 also shows dependencies of the form $k_{\mathrm{meas}} \propto u_{*,\mathrm{w}}^x$ with a variable exponent $x$. Clearly, the functional dependency between $k_{\mathrm{meas}}$ and $u_{*,\mathrm{w}}$ dramatically changes at a friction velocity of around $6\,\mathrm{cm\,s^{-1}}$, indicating a new regime starting at this

**Figure 6.** Measured gas transfer velocities $k_{\text{meas}}$ in fresh water in Kyoto (a), in the seawater model in Kyoto (b) and in seawater in Miami (c) and (d). Tracers in the legend are sorted by increasing solubility. As a visual guide, lines showing exponential proportionalities between the transfer velocity and the friction velocities to varying powers are shown.





friction velocity. This finding is in good agreement with the earlier measurements of Iwano et al. (2013) and Krall and Jähne (2014), who also found a transition to a much steeper increase at $33\,\mathrm{m\,s^{-1}}$. Also, this wind speed coincides with the change in the $u_*(u_{10})$ relationship discussed in section 4.1.

A closer look at the fresh water transfer velocities (Fig. 6a) reveals an unexpected result. Even for high wind speeds all

tracers (except He and MA) have transfer velocities within a very narrow band, even though their solubility differs by several orders of magnitude. This is a clear indication, that transfer through closed bubble surfaces is much slower than the transfer through the water surface for fresh water even at the highest wind speeds.

In seawater and in simulated seawater (Figs. 6b and 6c), a clear spacing between the transfer velocities of tracers with different solubilities at high wind speeds can be seen. This means that bubble-induced gas transfer does play a role for seawater.

## 4.4 Separation of gas transfer across the free surface and bubbles

Once bubbles influence air-sea gas transfer, a separation of the different contributing mechanisms is required. Because of the additional influence of solubility, it is not possible to simply apply Schmidt number scaling. This is why the model combining gas transfer across the free water surface and bubble surface was developed in Sec. 2.2, see Eq. 10. Because all measurements were made at high wind speeds with clean water, the Schmidt number exponent was fixed to 0.5. Then, three unknown

parameters remain for each measuring condition:

- The transfer velocity across the free water surface at a Schmidt number of 600, $k_{\mathrm{s},600}$

- The limiting or maximum transfer velocity across the closed-bubble surfaces at a Schmidt number of 600, $k_{\mathrm{c},600}$. It is reached for gases with a solubility $\alpha \ll \alpha_t$.

- The transfer velocity associated with the bubble volume flux per water surface area, $k_{\mathrm{r}}$. In the limit of high solubilities (

$\alpha \gg \alpha_t$) the bubble-mediated gas transfer velocity is $k_{\mathrm{r}}/\alpha$, compare Eq. 7.

Because of the multi-tracer approach with more than three tracers covering a wide range of solubilities, it is possible to retrieve all three parameters of the model (Eq. 10) for each measuring condition separately and thus to separate the gas transfer across the free water surface from the bubble-induced gas transfer. In addition the transition solubility $\alpha_t$ can be computed according to Eq. 8. The model equation 10 was fitted to the data using a least squares algorithm with the free parameters

$k_{\mathrm{s},600}$, $k_r$ and $k_{\mathrm{c},600}$. MA was excluded from the fit due to its additional air side resistance. Also, He had to be excluded. Including He led to unrealistically low transition solubility of below 0.001. One possible reason for this is the high diffusion coefficient of He, and the resulting fast gas transfer from water into the bubbles, which might deplete the He concentration in the water between the bubbles inside bubble clouds. This effect has been observed and described before by Woolf et al. (2007). Another explanation is spray-induced gas transfer, which could limit the gas transfer velocity of tracers with high diffusivity

as discussed in Sec. 2.3.

Also, as a further quality criterion, the fit was required to obey

$$k_{c,600} \leq k_{\mathrm{meas,T}} \left( \frac{Sc_T}{600} \right)^{0.5} - k_{s,600} \tag{16}$$





**Figure 7.** Fitted contribution of the different components to the gas transfer velocity: a) surface transfer velocity $k_{s,600}$, b) bubble surface transfer velocity $k_{c,600}$, and c) the bubble volume $k_r$ as a function of the water side friction velocity in double-logarithmic presentation. Please note the different vertical scales. The graphs include error bars of the fitted parameters. In addition the transition solubility $\alpha_t$ computed according to Eq. 8 in shown in figure d) without error bars.





by limiting the available parameter space of $k_{c,600}$ for tracers $T$ with solubilities smaller than 0.01 and Schmidt numbers larger than 300 (i.e. for the tracers SF$_6$ and CF$_4$). Eq. 16 describes the highest physically reasonable $k_{c,600}$ (see also Eq. 11). At each measuring condition the regression with three free parameters was performed with 5–14 measured transfer velocities. For some tracers, the concentrations of two different ions of the same tracer were analyzed with the MIMS, which allowed the

measurement of two transfer velocities per tracer for a single wind speed condition, see Sec. 3.3.

The mean (median) deviation between the measured and the modeled transfer velocity is 7.4 % (6.3 %) of the measured transfer velocity. Out of a total of 242 pairs of measured and modeled values, only 22 deviate by more than 15 %. The maximum deviation found was 31.2 % (Acetylene in seawater at $u_{10} = 58.6\,\mathrm{m\,s^{-1}}$). This indicates that the regression model is in good agreement with the measured data.

Figure 7 shows the resulting fitted parameters $k_{s,600}$, $k_r$ and $k_{c,600}$ and the calculated transition solubility $\alpha_t$ (see Eq. 8). The bubble-related parameters $k_r$ and $k_{c,600}$ are found to be zero for friction velocities below $5.8\,\mathrm{cm\,s^{-1}}$ for fresh water and seawater. No experiments below $7\,\mathrm{cm\,s^{-1}}$ were performed for the seawater model. The separation of the gas transfer velocity into its different components gives a detailed insight into the mechanisms of air-sea gas transfer at high wind speeds with unexpected results:

**Free surface transfer, $k_{s,600}$:** The gas transfer velocity across the free water surface $k_{s,600}$ normalized to a Schmidt number of 600 (Fig. 7(a)) clearly shows a transition to a much steeper increase of the transfer velocity with the friction velocity from $\propto u_{*,\mathrm{w}}^1$ to $\propto u_{*,\mathrm{w}}^{3.52}$ beyond a friction velocity of about $5.8\,\mathrm{cm\,s^{-1}}$. It is a substantial effect, resulting in a more than tenfold gas transfer velocity if the water side friction velocity is increased by a factor of two from 6 to $12\,\mathrm{cm\,s^{-1}}$. This substantial increase of the gas transfer velocity is not related to bubble-induced gas transfer at all and thus valid also

for all water-side controlled gas tracers independent of the solubility. It is not unexpected that there is no significant difference between seawater and fresh water, because the hydrodynamic conditions do not depend on the salt content of the water and the normalization of the transfer velocity to a Schmidt number of 600 already takes the small change of the kinematic viscosity between seawater and fresh water into account. It is more surprising that there is no significant difference between the Kyoto and SUSTAIN facilities although they differ significantly in lengths (15.7 m versus 24 m)

and width (0.8 versus 6 m).

**Bubble surface related transfer, $k_{c,600}$:** Bubble-induced gas transfer could only be observed after the transition to a much steeper increase of the gas transfer velocity at the surface beyond a friction velocity of $5.8\,\mathrm{cm\,s^{-1}}$. (Fig. 7(b) and (c)). The maximum bubble-induced gas transfer velocity in the limit of low solubility $k_{c,600}$ increases even steeper (Fig. 7(b)). For all fresh water conditions studied, bubble-induced gas transfer remains much smaller than the gas transfer at the free

water surface. For seawater, however, $k_{c,600}$ is an order of magnitude higher and surpasses $k_{s,600}$ at a friction velocity of about $8\,\mathrm{cm\,s^{-1}}$. At the highest wind speed it is about 1.7 times larger than at the free surface. This is shown in Fig. 8, where the ratio of $k_{c,600}$ to $k_{s,600}$ is shown as a function of the friction velocity. The about tenfold larger bubble-induced gas transfer velocity for seawater than for fresh water in the limit of low solubilities is in good agreement with the measured bubble surface as presented in Sec. 4.2.





**Figure 8.** Ratio of $k_{c,600}$ to $k_{s,600}$, i. e., the bubble-induced gas transfer velocity in the limit of low solubility to the gas transfer velocity at the free surface in relation to the friction velocity, given in percent of the gas transfer velocity at the free surface.



**Bubble volume flux density related transfer, $k_r$:** In the limit of high solubility, bubble-induced gas transfer is not significant at all (Fig. 7(c)). It is also not unexpected that there is no significant difference between seawater and fresh water, because in this limit, bubble-induced gas exchange is controlled by the bubble volume flux. It can be expected that the gas volume submerged per breaking wave does not depend on the salt content, because this depends only on the geometry and dynamics of wave breaking. Interestingly, Mischler (2014) found $k_r = 40\,\mathrm{cm\,h^{-1}}$ in his bubble tank study, a value close to the values at highest wind speeds.

**Transition solubility $\alpha_t$:** In seawater, however, many more small bubbles are generated, which stay longer in the water and form a significantly larger surface. This is why seawater is much more effective in bubble-induced gas transfer for low solubilities. Therefore also the transition from surface to volume flux controlled bubble-mediated gas transfer is shifted for seawater to lower solubilities (Fig. 7(d)) from fresh water at about 0.4 of values around 0.03. Within the measurement accuracy, no difference was found between seawater and simulated seawater. Thus not only the absolute values of bubble-induced gas transfer (Fig. 7(b) and (c)) but also the transition solubility is correctly reproduced when using traces of n-Butanol in fresh water to simulate the effect of seawater on bubble generation and its effects on air-sea gas transfer. This greatly simplifies laboratory experiments.

In his bubble tank experiment Mischler (2014) found very similar transition values: for fresh water 0.23 and for saltwater 0.06 at the conditions shown in Fig. 2. The deviations are not surprising, because in a bubble-tank without wind, where the breaking waves are simulated by a jet, the turbulence in the water certainly is different.

The successful partitioning of the gas exchange according to Eq. 10 makes it possible to compute the gas transfer velocity for any gas. This is important because it is now possible to determine the transfer velocity of other important species, which could not be measured with our setup, e. g., carbon dioxide.

### 4.5 Dimethyl sulfide and carbon dioxide

The gas transfer velocities measured for dimethyl sulfide (DMS) deserve some more discussion here, because they partly contradict the results of previous field experiments.

The eddy covariance measurements of Bell et al. (2013, 2015) found a decrease of the gas transfer for wind speeds higher than approx. $15\,\mathrm{m\,s^{-1}}$ (Fig. 9). The measurements presented here do not show such an effect (Fig. 9). On the contrary: beyond a wind speed of about $33\,\mathrm{m\,s^{-1}}$, the transfer velocity shows the same transition to a much steeper increase as all other tracers used. More recent field measurements do not show a decrease in the DMS transfer velocity (Blomquist et al., 2017). Zavarsky et al. (2018) found that the increase in the gas transfer velocity with wind speed becomes less steep at higher wind speeds.

Because it is well known that parameters other than wind speed influence gas exchange (Jähne, 2019), a direct comparison of field data with laboratory data based on the wind speed alone is not adequate. Therefore, the remaining question is whether it is physically reasonable that in a wind speed range of 12 to $20\,\mathrm{m\,s^{-1}}$ the gas transfer velocity varies by almost a factor of three at the same wind. Currently there is no evidence available from laboratory measurements that would confirm this. The influence of fetch and thus waves on gas transfer has so far only been studied at wind speeds of less than $10\,\mathrm{m\,s^{-1}}$ and proved



**Figure 9.** Measured transfer velocities of DMS, scaled to $k_{600}$, compared to previous field studies. B2013: Bell et al. (2013); B2015: Bell et al. (2015); B2017: Blomquist et al. (2017); Z2018: Zavarsky et al. (2018).







**Figure 10.** Comparision of DMS and carbon dioxide gas transfer velocities in a double logarithmic representation: eddy covariance measurements from the High Wind Speed Gas Exchange Field Study (HiWinGS) Blomquist et al. (2017) (B2017). Also shown are the $CO_2$ and DMS transfer velocities measured by Zavarsky et al. (2018) (Z2018). The output of the model presented in this paper for $CO_2$ and DMS is also shown.

to be only significant at wind speeds lower than $6\,\mathrm{m\,s^{-1}}$ (Kunz and Jähne, 2018). However, recent measurements in the Baltic Sea using active thermography also showed large variations in the gas transfer velocity at wind speeds higher than $10\,\mathrm{m\,s^{-1}}$, which is most likely related to the effect of surfactants (Nagel et al., 2019).





There is another striking discrepancy. Blomquist et al. (2017) found significantly higher gas transfer velocities for carbon dioxide than for DMS and attributed this to bubble-induced gas transfer, whereas the measurements presented here do not show any significant bubble-induced gas exchange for both gases in the wind speed range covered by the HiWinGS study and even at much higher wind speeds (Fig. 10). Actually, the figure shows that the transfer velocity for carbon dioxide is an almost

constant factor of three higher than that of DMS in the whole wind speed range from 3.4 to $25\,\mathrm{m\,s^{-1}}$ for the field data set of Blomquist et al. (2017). It is unlikely that a) already at $3.4\,\mathrm{m\,s^{-1}}$ wind speed, the bubble-induced gas exchange increases the transfer velocity of carbon dioxide threefold over DMS and b) that this effect neither increases, nor decreases with wind speed. Also, no such discrepancy between the gas transfer velocities of DMS and $CO_2$ spanning all wind speeds was found in a similar study by Zavarsky et al. (2018), see Fig. 10. The laboratory measurements presented here may provide some insight

into the cause for this discrepancy.

Wave breaking in the ocean could be more intense than in a short-fetch wind-wave tank and thus also the bubble-induced gas exchange would be stronger and would start to become significant at lower wind speeds. Indeed, the bubble surface area measurements reported in Sec. 4.4 and Fig. 5b show larger bubble surface areas with increasing fetch at the same wind speed. In the Heidelberg Aeolotron with infinite fetch, the same bubble surface area occurs at about half the wind speed than in the

short-fetch Kyoto facility. However, even if the fetch effect was larger at the open ocean, bubble-induced gas exchange would become significant only beyond a critical wind speed, which is for sure larger than a calm $3.4\,\mathrm{m\,s^{-1}}$.

Su and Cartmill (1995) measured bubble distributions and void fractions in a 90 m long, 3.36 m deep and 3.66 m wide wave channel with large mechanically generated breaking waves using fresh water and artificial seawater. They found an about tenfold larger bubble surface area in sea water than in fresh water, but no significant change in the void fraction, which agrees

well with our findings. This is a strong indication that the fundamental physical mechanisms for bubble-induced gas exchange are not sensitive to the scales from rather short and shallow wind-wave tanks used in this study to longer and deeper wave tanks. Therefore, it is unlikely that the physical mechanisms are different in the ocean, and the transition solubility $\alpha_t$ in the field is expected to be similar to $\alpha_t$ measured in the lab. In seawater it is around 0.03, which results in a reduction of bubble-induced gas transfer of carbon dioxide by a factor of more than 20 in comparison to the low-solubility limit $k_{c,600}$, see Eqns. 7, 8 and

Fig. 10. Consequently, bubble-induced gas transfer is not significant for both carbon dioxide and DMS and therefore cannot be the cause for the different gas transfer velocities reported in Blomquist et al. (2017).

### 4.6 Comparison with field data at hurricane wind speeds

As a final step, Fig. 11 shows a comparison of our results with the only available field data set at hurricane wind speeds (McNeil and D'Asaro, 2007). The oxygen gas transfer velocities were calculated according to Eq. 10 and the model equations are shown

in App. A. It is interesting to see how close our wind-wave tank data are to the field data. This does not mean that the laboratory data can simply be extrapolated to the field. This would physically not be correct and in addition, the uncertainties of the field data are too large for such a statement. It does show, however, that wind-wave tank studies do not miss an essential mechanisms compared to ocean conditions.





**Figure 11.** Comparison of oxygen gas transfer velocities inferred from the lab measurements presented in this paper with the field data by McNeil and D'Asaro (2007), both Schmidt number scaled to $k_{600}$.





## 5  Conclusions

With multi-tracer gas exchange experiments in two high-speed wind-wave tanks it was possible to separate the mechanisms of air-sea gas transfer into its different components. In the short-fetch tanks, a steep increase of the transfer velocity across the free surface was found beyond wind speeds of $33\,\mathrm{m\,s^{-1}}$ (friction velocity in water $5.8\,\mathrm{cm\,s^{-1}}$) increasing the transfer velocity

corrected to a Schmidt number of 600 from $110\,\mathrm{cm\,h^{-1}}$ to a maximum measured value of about $1600\,\mathrm{cm\,h^{-1}}$. This part of the gas transfer is the same in fresh water and seawater.

It is obvious that a new regime is established at wind speeds beyond $33\,\mathrm{m\,s^{-1}}$, which is governed by the intense turbulent mixing and permanent rapid disruption of the surface. The detailed mechanisms causing the steep increase of the gas transfer velocity at high wind speeds are still unclear and require further investigations. Because this effect is clearly not caused by gas

transfer through closed bubble surfaces, it can be explained as either significantly enhanced turbulence at the water surface, or a significantly enlarged surface area for the exchange processes, or a combination of both. Many processes must be considered at highest wind speeds, including the generation of steep small-scale surface waves, the fragmentation of wave crests where the bag-breakup mechanism is dominant (Troitskaya et al., 2017), the effects of high-speed spray and spume droplets plunging into the water surface again and the effects of bursting bubbles. The finding of the relatively low transfer velocity for He at the

highest wind speed (Sec. 4.4) is a first indication that rapid surface fragmentation processes play an important role, but further studies are required.

In fresh water bubble-induced gas transfer does not play a significant role at all. Even at the highest wind speed and in the high limit for low soluble gases it is just about 25% of the gas transfer across the free water surface. In seawater bubble-induced gas transfer is about an order of magnitude larger and becomes an important contribution for gases with low solubility.

At the highest measured wind speed of $80\,\mathrm{m\,s^{-1}}$ it is about 1.7 times larger than the gas transfer at the free water surface. For moderately soluble gases such as the widely studied tracers carbon dioxide and DMS, bubble-induced gas transfer is still not a significant process because transition solubility $\alpha_t$ from the surface-related to volume flux related gas transfer was found to be at a quite low value of around 0.03. Therefore, DMS and carbon dioxide should show the same gas exchange velocities at all wind speeds.

Bubble measurements in two additional facilities, especially in the annular Heidelberg Aeolotron, suggest that the steep increase of bubble concentrations is likely shifted to lower wind speeds at infinite fetch. This means that the high wind speed regime could start at lower wind speeds for larger fetches. However, the effect cannot be too large because the short-fetch wind-wave tank results agree surprisingly well with the only field data set at hurricane wind speeds by McNeil and D'Asaro (2007).

In field experiments it remains very difficult to reveal the mechanisms of air-sea gas transfer because there are not enough tracers available to span the necessary wide range of tracer solubility and diffusivity. Systematic field studies addressing the mechanisms are therefore hardly possible. As this study has shown, systematic and well-designed wind-wave tank experiments have more potential to reveal the mechanisms of the gas transfer processes. This opens also the opportunity to predict transfer velocities under field conditions.





However, the most serious limitation is the short fetch of the linear laboratory facilities. High wind speed gas transfer studies in the annular Heidelberg Aeolotron with infinite fetch have the potential to close the "fetch gap" between the laboratory and the field.

*Data availability.* All measured data reported and discussed in this paper will be published on the free and open digital archive zenodo within the small-scale air-sea interaction community, https://zenodo.org/communities/asi once this paper has been accepted for publication. All third party data sets used are cited in the text.

## Appendix A: Model equations

All $u_{*,w}$ in $\mathrm{cm\,s^{-1}}$, all $k$ in $\mathrm{cm\,h^{-1}}$.

$k_{s,600}$ for fresh water and seawater:

$$k_{s,600} = \begin{cases} 8.225^{-1}\ u_{*,w}\ 600^{-1/2} & \text{if } 0.75 < u_{*,w} < 5.8 \\ 0.214\ u_{*,w}^{3.52} & \text{if } 5.8 \leq u_{*,w} < 13 \end{cases} \tag{A1}$$

$k_{c,600}$ for seawater:

$$k_{c,600} = \begin{cases} 0 & \text{if } 0.75 < u_{*,w} < 5.8 \\ 4.01(u_{*,w} - 5.8)^{2.20} & \text{if } 5.8 \leq u_{*,w} < 13 \end{cases} \tag{A2}$$

$k_{c,600}$ for fresh water:

$$k_{c,600} = \begin{cases} 0 & \text{if } 0.75 < u_{*,w} < 5.8 \\ 51.3(u_{*,w} - 5.8)^{2.07} & \text{if } 5.8 \leq u_{*,w} < 13 \end{cases} \tag{A3}$$

$k_r$ for fresh water and seawater:

$$k_r = \begin{cases} 0 & \text{if } 0.75 < u_{*,w} < 5.8 \\ 1.19(u_{*,w} - 5.8)^{2.044} & \text{if } 5.8 \leq u_{*,w} < 13 \end{cases} \tag{A4}$$

*Author contributions.* KEK planned the experiments, performed the measurements, evaluated the data and prepared all figures. BJ contributed to planning of the experiments, worked on the bubble model and drafted the main conclusions. Both authors contributed equally to writing the paper.

*Competing interests.* None





*Acknowledgements.* The authors gratefully acknowledge the significant help of many people. Satoru Komori and Brian Haus kindly allowed us to run their wind-wave tanks at the most extreme wind speeds. Wind data in Miami was provided by Andrew Smith. Wolfgang Mischler and Angelika Klein contributed to the measurements in Kyoto. Sonja Friman and Jan Bug participated in the measurements in Miami. Naohisa Takagaki, Andrew Smith, Michael Rebozo, Cedric Guigand, Neil Williams, Nathan Laxague, David Ortiz-Suslow and Brian Haus

5   helped with the set-up of our equipment and provided invaluable logistical support. We gratefully acknowledge partial financial support of this research by the German Science Foundation (DFG), grants JA 395/17-1&2 "Air-Sea Gas Exchange at High Wind Speeds".





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
