# Peer review of "Air-sea gas exchange at wind speeds up to 85 m $\rm s^{-1}$"

_Ocean Science, 2019_

## Referee Comment (RC1) · Byron Blomquist (Referee) · 16 Aug 2019

This study presents results from an extensive series of air-water gas transfer experiments in laboratory wind-wave tanks with fresh water and seawater, utilizing several gases covering a wide solubility range. The focus is on high wind speed conditions. The principle merit of gas transfer studies in a wind-wave tank is the ability to precisely determine gas flux by measuring the loss of dissolved gas in the liquid phase over time. The use of gases with widely differing solubility is another strength of this study, permitting an assessment of the interfacial and bubble-mediated gas transfer mechanisms.

The manuscript is very well written and generally clear. Some points requiring more

explanation will be mentioned below. There are very few problems with usage, spelling or punctuation so I will confine these remarks to the experimental results and analysis.

The experimental set up and methodology are clearly described. However, I would like to see a bit more detail on the assumptions involved in deriving an open ocean-equivalent 10m wind speed and u* from wind speed measurements in the wind tunnel (p.12). This is important for judging the comparison to field measurements.

I don't fully understand how the parameters defined on p.16 (ks600, kc600 and kr) were obtained from the measurements. Was kc600 determined using only data for SF6 and CF4 (and only SF6 in seawater), as mentioned on p.18 and are these results shown in Fig.7b? Were these then applied as fixed values in a two-parameter fit to data for all gases to obtain ks600 and kr in Fig. 7a,c?

There's potential confusion with the notation for kc600, defined on P.16 as a constant (maximum) value for bubble transfer at a given u*, because kc is also the second term in Eq.10 which could be measured under conditions where Sc=600, but would not be the same as kc600 defined on p.16 since it depends on gas solubility. I suggest using a different notation for the fit parameter representing the maximum limiting value of kc. Perhaps results for kc can also be shown on a plot similar to Fig.2, where the 'kc600' parameter is indicated as the value of kc at the low solubility limit, where the curve is flat?

I'm surprised the authors do not present a detailed comparison with results from Rhee et al. 2007, which is a similar wind-wave tank gas transfer study and should be more directly comparable to this work than the field studies.

The absence of detectable bubble transfer below u*w=5.8 m/s for all gases is certainly unexpected, and to me a sign that something is very wrong here. For example, from the information presented in Fig.2 (Mischler, 2014) we expect kc for CO2 (alpha=0.78 @ 20°C) and kc for DMS (alpha=12 @ 20°C) to differ by more than a factor of 10. The absence of any difference in transfer rate at moderately high wind speeds among

gases covering a broad solubility range is an indication that something is wrong in the determination of kc, or that the experimental design is unable to simulate mechanisms of gas transfer at these wind speeds at sea. This result is certainly contradicted by field evidence from several studies showing a generally linear increase in k for DMS at wind speeds of 10-20 m/s and a roughly quadratic increase for a less soluble gas like $CO_2$ over the same interval.

I don't see obvious errors in the theoretical model developed by the authors, which is generally similar to prior treatments in the literature. I suspect the unique conditions in the wind-wave tank at high wind speeds are not comparable to the open ocean. Even an 'infinite fetch' design cannot simulate the wave spectrum in open ocean conditions, except perhaps under light winds, and thus cannot simulate large breaking wave crests and deep bubble plume penetration. I therefore wonder if the absence of bubble-mediated transfer at moderate wind speeds and the observed abrupt jump in the slope of gas transfer at wind speeds above 30 m/s are merely characteristics inherent to the wind-wave tank experimental design?

I assume high wind interfacial conditions in the tank to correspond to a 'young' sea state, with very high surface stress and widespread coverage with small, choppy breaking waves. This condition is not common at sea except in a situation of very short fetch or a very rapid increase in wind speed, and in any case does not persist long before large breaking waves develop. It's therefore difficult to understand how these results apply to typical 'hurricane wind speed' conditions at sea. The authors should present a detailed analysis of these differences to provide some context for comparisons with field studies.

DMS is the high-solubility gas in this study (MA was omitted) and should represent interfacial transfer with minimal bubble-mediated contribution. The comparison to data from field studies in Fig.9 looks fairly good to me, despite the fact that there is little or no overlap in the wind speeds. Thus results for the first term in Eq.10, ks, seem roughly consistent with open ocean observations. Instances of suppressed DMS transfer noted

in a few field studies are the exception and suggest we don't yet understand all the factors controlling gas transfer. The effects of surfactants are an obvious factor that probably suppresses gas transfer, with some support from lab studies, but this has not been carefully examined under field conditions except at low wind speeds. Zavarsky et al. 2018 discuss the possible suppression of transfer by flow separation and angular differences in wind and wave direction.

With respect to the comparison with results in B2017 (Fig.10 and p.23), I can make a few clarifications. The B2017 cruise focused on high wind conditions with relatively few flux measurements at U10< 8 m/s, and these are generally under non-ideal conditions when the ship was moving at maximum cruise speed to reposition between storm events. So, we expect additional uncertainty or bias in the low wind speed results. Trends shown by the bin averages in Fig.10 are therefore misleading, and in any case the error bars for kdms and kco2 overlap at low wind speeds, so it's not meaningful to say results for the two gases differ by a factor of 3 at U10=3.4 m/s.

Nevertheless, at moderate wind speeds of 10-16 m/s sampled under ideal conditions, kco2 from B2017 shows quite a bit of scatter and a high bias compared to other studies, with lower transfer rates observed in 'young' sea states and enhanced transfer in fully developed conditions or in 'old' seas when wind speed is declining but waves are still quite large. These effects are less pronounced for DMS. See Fig.6 in B2017. This implies sea state is a significant factor in the transfer of low solubility gases, and these subtleties are obscured by bin averaging. The comparison between kdms and kco2 likely depends on the specific sea state conditions, and the bubble transfer contribution to low solubility gases in a very 'young' sea state may be significantly reduced, which could be consistent with the kc result in this report.

I think this is a carefully conducted study and well written report which explores the mechanisms of gas transfer in a wind-wave tank, but I struggle to understand the significance of these results with respect to conditions in the open ocean, especially at 'hurricane wind speeds'. I don't agree with the conclusion in Sec.4.6 that rough correspondence between the wave-tank and open ocean data in Fig.11 shows the lab results are capturing the essential mechanisms, since the mechanistic details in each case could be significantly different (the physical details certainly are) and the rough agreement coincidental. As someone with a keen interest in this topic but limited experience with of wind-wave tank experiments I'd like to see a more thorough examination of these issues.

Rhee, T. S., P. D. Nightingale, D. K. Woolf, G. Caulliez, P. Bowyer, and M. O. Andreae (2007), Influence of energetic wind and waves on gas transfer in a large wind–wave tunnel facility, J. Geophys. Res., 112, C05027, doi:10.1029/2005JC003358.

---

## Referee Comment (RC2) · Christopher Fairall (Referee) · 21 Aug 2019

This paper is a description of an analysis of water- wind tunnel measurements of the rate of transfer of trace gases. Twelve gases are uses and the simulations are for very strong winds – crudely equivalent to 10-m atmospheric wind speeds up to 80 m/s. The results are expressed as a parameterization of water-side gas transfer velocity, kw, in terms of the water-side friction velocity, u*w. The authors use the breadth of the solubility of the gases to separate the so-called direct or free surface (air-water molecular scale) transfer to two bubble mediated mechanisms which they refer to a 'surface area' and 'volume flux' mechanisms. Their main finding – that bubble transfer is negligible compared to the direct transfer for all gases up to wind speeds of 30 m/s – is surprising (at least to this reviewer). For example, the COARE gas transfer

algorithm (Fairall et al. 2011; Blomquist et al. 2017), which is based on Woolf's 1997 bubble parameterization gives bubble and direct transfer approximately equal at a wind speed of 8-10 m/s. The authors discuss this in their section 4.6 and conclude the difference in CO2 and DMS transfer velocity observed by Blomquist et al cannot be due to bubbles. Certainly an amusing controversy that is worth airing. The paper is fairly well written and the authors are certainly experts in the field. They try conscientiously to go the extra mile in explaining the differences in free surface and bubble transfer mechanisms and even provide information on possible sea spray transfer (although that cannot be assessed with their measurements). The paper is, with a few exceptions, well referenced. While the overall implications for open ocean gas transfer are open to interpretation, the actual wind tunnel measurements appear to be solid. So, in my view the paper should be published with some clarifications and improvements. The authors' conclusions are difficult to reconcile with open ocean measurements but I leave it to them to consider how to handle that. In my opinion they are too dismissive of the large number of experimental (tracers and eddy covariance) studies that indicate insoluble gases and CO2 have substantially higher transfer velocity above 15 m/s. I question that the Zavarsky paper is sufficient cover.

I think the paper would benefit from more careful consideration of mixing actual measurements, assumptions, and inferences. I don't understand why they used Powell's open ocean estimates of Cd when they could have used Donelan's 2004 results actually determined in the Miami wind tunnel. Also, please give us a sentence explaining Takagadi's method for getting friction velocity so we don't have to go look it up. I am guessing they assumed a momentum balance at the interface to compute u*w from u*a (square root of ratio of air to water density). This assumes that the growth of the wave field has negligible effect of the momentum balance. Is this right? Also, they switch back and forth between u*w and just u* - I assume they mean the same thing. Just be advised that it is quite a stretch from a setting on an instrument dial to actual waterside friction velocity. Another example, McNeil and D'Asaro (2007) did not 'measure' gas transfer velocities, they inferred them from water concentration measurements. Furthermore, their basic assumption in the analysis is that both free surface and bubble transfer velocities scale with u*w and nothing else. However, there is considerable evidence that the air volume flux from breaking scales as u* and other wave parameters (see Deike et al. 2017). Also note that Deike and Melville (2018) used this approach to estimate kb for DMS and $CO_2$ – treating both gases as highly soluble. They present measurements of bubble area from the wind tunnels but no estimates of volume flux. I think the bubble volume flux data should appear in Fig. 7c. Finally, the authors might note that Rhee et al. 2007 found considerable enhancement of k for insoluble gases when bubbles were introduced.

Page 2 line 19. U*a is a measure for momentum extracted from the wind. Some of that momentum is realized in the ocean via direct viscous transfer at the interface, some goes into growing waves and some is later realized as turbulence when waves break. If locally waves are in dynamic balance, then the momentum flux from the air is the same as the momentum flux realized in the ocean. So, how close is the balance in a wind tunnel?

Page 3 line 12. Bubbles may also suppress turbulence through density stratification.

Eq (5) This terminology is confusing with the un-numbered equation on Page 2 line 14.

Page 4 line 1 Suggest referencing bubble model work of Liang et al. GLOBAL BIO-GEOCHEMICAL CYCLES, VOL. 27, 894–905, doi:10.1002/gbc.20080, 2013 and earlier work.

Page 5 Eq (7) I am confused by the terminology. Can Qb be volume flux and Qb/As also be a volume flux? If we equate kr with the volume of air ingested per unit area per unit time (units velocity), then that should be on the order of 30 cm/hr at u10=15 m/s and u*w=2 cm/s (see, Deike et al, 2017). That does not compare well with Fig. 7 c, where kr doesn't reach those values until u*w is greater than 10.

Page 5 Eq (10) This equation is similar to Woolf parameterization. For the volume flux

is kr=2450*whitecap fraction (cm/h) and the parameter e=kc/kr*sqrt(600). How does this compare to your results?

Page 7 section 2.3. This discussion of droplet effects is a little confusing. It seems to me that ejecting a droplet does not change the waterside concentration, so their measurement method does not capture it. If the drop has time, it would transfer gas to the air and that would reduce free surface transfer further down the line. Is that what they are trying to say? This argument about time scales ignores the fact that the droplets leave the wind tunnel before they can do much transferring – this is discussed in Andreas and Mahrt 2016.

The discussion of field measurements of k for DMS and CO2 is very useful (Section 4.5). It also illustrates the rather inconclusive state of field observations. The data from Zavarsky et al (2018) show essentially no difference between CO2 and DMS. The analysis of Fairall et al. (2011) which compiled all the direct flux data to date showed significant differences. The HIWINGS data shown in this figure are quite surprising for CO2. Blomquist gives k CO2 a power dependence of u10^1.68, which is not linear. Because of the conditions, I don't think the HIWINGS data below U10 of 10 m/s should be considered. Even ancient information such as Wanninkhoff's famous formula indicate a quadratic wind speed dependence for insoluble gases. Earlier suggestions that k CO2 should scale as U^3 were based on the assumption that whitecap fraction scaled as U^3. More recent observations have shown that this is not the case, with much weaker wind speed dependence at high wind speeds. I think the authors may be placing too much importance on Zavarsky. From Wanninkhoff's radioactive tracers, to a number of deliberate tracer studies, and perhaps all other eddy flux measurements CO2 goes at least quadratic with wind speed. At u10=18 m/s, the value is close to 100 cm/h for open ocean measurements.

Figure 10. What drag coefficient is used for the curve shown for modeled DMS and CO2?

[Figure]

---

## Author Comment (AC1) · 3 Sep 2019

**by Byron Blomquist**

August 29, 2019

The authors thank Mr. Blomquist for his thorough and helpful comments. A point by point answer to his comments can be found below.

**Computation of $u_{10}$**

> Reviewer's comment: However, I would like to see a bit more detail on the assumptions involved in deriving an open ocean-equivalent 10 m wind speed and u* from wind speed measurements in the wind tunnel(p.12).

We now explain this in more detail in the manuscript. To this end we will add our co-workers from Kyoto (Naohisa Takagaki) and Miami (Andrew Smith) as co-authors, because they made the measurements and greatly helped with computing $u_{10}$. We also found a mistake in converting between $u_{*,a}$ and $u_{*,w}$, and decided to use Donelan et al. (2004) (lab measurements of $C_D$) instead of Powell et al. (2003) (field measurements of $C_D$) to convert measured wind speeds in Miami to $u_*$ and $u_{10}$. This slightly changes the relationships $k_x(u_{*,w})$ and the model parameterization equations, but not our findings. Further details and an in-depth description of the measurement procedure can be found in Takagaki et al. (2012) and Donelan et al. (2004).

**Separation of total gas transfer velocity into the components used in Eq. 10.**

> Reviewer's comment: I don't fully understand how the parameters defined on p.16 (ks600, kc600 and kr) were obtained from the measurements. Was kc600 determined using only data for SF6 and CF4 (and only SF6 in seawater), as mentioned on p.18 and are these results shown in Fig.7b? Were these then applied as fixed values in a two- parameter fit to data for all gases to obtain ks600 and kr in Fig. 7a,c?

Each wind speed is treated separately. The fit routine does not know $u_*$ or $u_{10}$. Input parameters are: all $k_{\text{meas}}$, of one wind speed condition, Sc and $\alpha$ calculated at the water temperature $k_{\text{meas}}$ was measured at. In a first step, the fit routine minimizes $(k_{\text{tot}}\text{-}k_{\text{meas}})^2$ using a standard least squares algorithm (`scipy.optimize.curve_fit` in python) where

- $k_{\text{meas}}$ is the set of all measured $k$ at one specific wind speed condition, and

- $k_{\text{tot}}$ is calculated from the corresponding physico-chemical tracer properties $\alpha$ and Sc using Eqn. 10 with the free and to be optimized parameters $k_{c600}$, $k_{s600}$ and $k_r$.

In the next step(s), the condition given by equation 16 is looked at. If it is fulfilled, the fit routine commences and outputs $k_{c600}$, $k_{s600}$ and $k_r$ from step 1. If the condition Eqn. 16 is not fulfilled, the fit is repeated, however the parameter space is reduced for $k_{c600}$, with the maximum allowed value of $k_{c600}$ being $k_{c,\text{max}} = k_{meas,T,600} - k_{s600}$ where $k_{meas,T,600}$ is the highest measured, Schmidt number scaled transfer

velocity of either $SF_6$ or $CF_4$. This second fit yields a new set of $k_{c600}$, $k_{s600}$ and $k_r$, for which the check according to equation 16 is performed again. This is repeated until the condition is satisfied, and the fit routine commences with the results $k_{c600}$, $k_{s600}$ and $k_r$ from the last iteration step.

This is repeated for each wind speed condition of each of the campaigns separately.

$k_c(\alpha)$-curves as well as plots showing a comparison between $k_{meas}$ and $k_{modeled}$ are shown in the appendix below.

**The parameter $k_{c,600}$**

> Reviewer's comment: There's potential confusion with the notation for kc600, defined on P.16 as a constant(maximum) value for bubble transfer at a given u*, because kc is also the second term in Eq.10 which could be measured under conditions where Sc=600, but would not be the same as kc600 defined on p.16 since it depends on gas solubility. I suggest using a different notation for the fit parameter representing the maximum limiting value of kc. Perhaps results for kc can also be shown on a plot similar to Fig.2, where the 'kc600' parameter is indicated as the value of kc at the low solubility limit, where the curve is flat?

The definitions are indeed consistent. Maybe this line of reasoning helps:
Starting from Eqn. 9,

$$k_c = \frac{1}{\alpha} k_r \left[ 1 - \exp\left( -\frac{\alpha}{\alpha_t} \right) \right],$$

assuming that the exponent $\alpha/\alpha_t$ is small, we can calculate the Taylor series up to the second term,

$$\exp\left( -\frac{\alpha}{\alpha_t} \right) = 1 - \frac{\alpha}{\alpha_t}.$$

Inserting this into Eqn. 9 above immediately cancels $\alpha$, so that $k_c$ indeed does no longer depend on $\alpha$ for small $\alpha$. Then, replacing $\alpha_t$ with its definition given in Eqn. 8 yields

$$k_{c,\text{low }\alpha} = k_{c,600} \left( \frac{600}{\text{Sc}} \right)^{n_b}$$

which is the definition of $k_{c,600}$ in the limit of low solubilities given in Eqn. 6 and also again given on P16. Also have a look at the $k_c(\alpha)$-curves in the appendix below, which show a flattening for the low solubilities. Maybe the confusion comes from the attempt to apply Schmidt number scaling to $k_c$ as given in Eqn. 9 for a gas with a solubility close to, equal or larger than $\alpha_t$ to arrive at something like a $CO_2$-equivalent bubble surface transfer velocity, which one might also be tempted to also call $k_{c,600}$. However, since $k_c$ as given in Eqn. 9 does depend on the solubility, Schmidt number scaling is not permitted, so that

$$k_c \left( \frac{Sc}{600} \right)^{-n} \neq k_{c,600}$$

for gases with a solubility close to, equal or larger than $\alpha_t$.

**Comparison with other wind-wave tank experiments**

> Reviewer's comment: I'm surprised the authors do not present a detailed comparison with results from Rhee et al. 2007, which is a similar wind-wave tank gas transfer study and should be more directly comparable to this work than the field studies.

Rhee et al. (2007) is, among other studies, rather irrelevant for our work, because 1) their highest measured wind speed is 13 m/s, and 2) their means of bubble generation (submerged aerators) is very different from ours (wave breaking induced bubbles only). Therefore such a comparison is not meaningful. In the introduction, we refer, of course, to the two previous lab studies in the Kyoto high wind speed facility: Iwano et al. (2013, 2014) and Krall and Jähne (2014).

**Comparison with Mischler's bubble tank experiments and difference between DMS and $CO_2$**

Reviewer's comment: The absence of detectable bubble transfer below u*w=5.8 m/s for all gases is certainly unexpected, and to me a sign that something is very wrong here. For example, from the information presented in Fig.2 (Mischler, 2014) we expect kc for CO2 (alpha=0.78@ 20°C) and kc for DMS (alpha=12 @ 20°C) to differ by more than a factor of 10.

Figure 1 shows the modeled $k_c$ for DMS and $CO_2$ in salt water:

[Figure]

Figure 1: Modeled $k_c$ for DMS and $CO_2$ in salt water

Both are, as expected, different by more than a factor of 10. However, since $k_c$ for DMS and $CO_2$ are very small compared to the surface transfer velocity, this difference can hardly be spotted in Fig. 10, where we show the total modeled k of DMS and $CO_2$. This finding is in perfect agreement with Mischler (2014), who measured pure bubble-induced gas exchange in a special bubble tank.

The reviewer could have easily produced a graph like this for any gas by using the model parameterization equations for $k_{c,600}$ and $k_r$ given in the appendix of the manuscript together with Eqns. 8 and 9 and computed the difference in $k_c$ for DMS and $CO_2$ for himself.

To summarize, when there are bubble effects, we do indeed see the correct spacing between DMS and $CO_2$. Nothing is wrong with our data or the fit.

**Missing bubble-induced gas exchange at moderatly high wind speeds**

Reviewer's comment: The absence of any difference in transfer rate at moderately high wind speeds among gases covering a broad solubility range is an indication that something is wrong in the determination of kc or that the experimental design is unable to simulate mechanisms of gas transfer at these wind speeds at sea. This result is certainly contradicted by field evidence from several studies showing a generally linear increase in k for DMS at wind speeds of 10-20 m/s and a roughly quadratic increase for a less soluble gas like CO2 over the same interval.

The reviewer's argument is only partially true. Field measurements show a rather confusing picture. While the results of Blomquist et al. (2017) show significantly higher gas transfer velocities for $CO_2$ than for DMS, the results of Zavarsky (2018) do not (Figure 2). Why is this the case and why are the DMS gas transfer velocities of Blomquist et al. (2017) almost a factor of two lower than those of Zavarsky (2018)? Also, why are gas transfer velocities measured using dual tracer techniques using the very low solubility tracers He and $SF_6$ (which translates to very large expected bubble contribution) generally much lower than $CO_2$ transfer velocities measured with eddy covariance (see the compilation of field measurements in Garbe et al. (2014, Fig. 2.10)), even at wind speeds as high as 15 m/s?

**Dependency on sea state respectively wave age**

*Reviewer's comment: I don't see obvious errors in the theoretical model developed by the authors, which is generally similar to prior treatments in the literature. I suspect the unique conditions in the wind-wave tank at high wind speeds are not comparable to the open ocean. Even an 'infinite fetch' design cannot simulate the wave spectrum in open ocean conditions, except perhaps under light winds, and thus cannot simulate large breaking wave crests and deep bubble plume penetration. I therefore wonder if the absence of bubble-mediated transfer at moderate wind speeds and the observed abrupt jump in the slope of gas transfer at wind speeds above 30 m/s are merely characteristics inherent to the wind- wave tank experimental design?*

*I assume high wind interfacial conditions in the tank to correspond to a 'young' sea state, with very high surface stress and widespread coverage with small, choppy breaking waves. This condition is not common at sea except in a situation of very short fetch or a very rapid increase in wind speed, and in any case does not persist long before large breaking waves develop. It's therefore difficult to understand how these results apply to typical 'hurricane wind speed' conditions at sea. The authors should present a detailed analysis of these differences to provide some context for comparisons with field studies.*

*and later . . .*

*Nevertheless, at moderate wind speeds of 10-16 m/s sampled under ideal conditions, kco2 from B2017 shows quite a bit of scatter and a high bias compared to other studies, with lower transfer rates observed in 'young' sea states and enhanced transfer in fully developed conditions or in 'old' seas when wind speed is declining but waves are still quite large. These effects are less pronounced for DMS. See Fig.6 in B2017. This implies sea state is a significant factor in the transfer of low solubility gases, and these subtleties are obscured by bin averaging. The comparison between kdms and kco2 likely depends on the specific sea state conditions, and the bubble transfer contribution to low solubility gases in a very 'young' sea state may be significantly reduced, which could be consistent with the kc result in this report.*

The authors agree with the reviewer that air-sea gas transfer is not only related to the wind speed, but that the sea state, especially the wave age must be considered as well. But, again, current field results are quite confusing. As the reviewer mentioned, Blomquist et al. (2017) found lower gas transfer velocities in 'young' sea states than in 'old' seas for carbon dioxide and attributed this to higher bubble contributions at older seas. This finding is in strong contrast to estimates of air entrainment due to breaking waves by Deike et al. (2017). They found that the air entrainment is much lower at high wave ages. The effect is large, air entrainment scales roughly with the inverse wave age.

Our short-fetch experiment add results for very young wave ages, where the contribution of bubbles is low again. Therefore currently the issue of wave age dependency needs to be left open. Systematic measurements covering a wide range of wave ages are required.

**DMS gas transfer**

*Reviewer's comment: DMS is the high-solubility gas in this study (MA was omitted) and should represent interfacial transfer with minimal bubble-mediated contribution. The comparison to data from field studies in Fig.9 looks fairly good to me, despite the fact that there is little or no overlap in the wind speeds. Thus results for the first term in Eq.10, ks, seem roughly consistent with open ocean observations. Instances of suppressed DMS transfer noted in a few field studies are the exception and suggest we don't yet understand all the factors controlling gas transfer emphasis added. The effects of surfactants are an obvious factor that probably suppresses gas transfer, with some support from lab studies, but this has not been carefully examined under field conditions except at low wind speeds. Zavarsky et al. 2018 discuss the possible suppression of transfer by flow separation and angular differences in wind and wave direction.*

*With respect to the comparison with results in B2017 (Fig.10 and p.23), I can make a few*

clarifications. The B2017 cruise focused on high wind conditions with relatively few flux measurements at U10¡ 8 m/s, and these are generally under non-ideal conditions when the ship was moving at maximum cruise speed to reposition between storm events. So, we expect additional uncertainty or bias in the low wind speed results. Trends shown by the bin averages in Fig.10 are therefore misleading, and in any case the error bars for kdms and kco2 overlap at low wind speeds, so it's not meaningful to say results for the two gases differ by a factor of 3 at U10=3.4 m/s.

[Figure]

Figure 2: Comparision of DMS and $CO_2$ gas transfer velocities in a double logarithmic representation: eddy covariance measurements from HiWinGS by Blomquist et al. (2017) (B2017). Also shown are the $CO_2$ and DMS transfer velocities measured by Zavarsky et al. (2018) (Z2018). The output of the model presented in this paper for $CO_2$ and DMS is also shown.

We thank the reviewer for this clarification. However, if you plot the individual measurements instead of bin-averaged measurements (Fig. 2), the transfer velocities of carbon dioxide are still significantly higher down to 3 m/s wind speed. We will use then individual measuring points in a revised version of Fig. 10 instead of bin-averaged values, see Fig. 2.

**Conclusions**

Reviewer's comment: I think this is a carefully conducted study and well written report which explores the mechanisms of gas transfer in a wind-wave tank, but I struggle to understand the significance of these results with respect to conditions in the open ocean, especially at 'hurricane wind speeds'.
I don't agree with the conclusion in Sec.4.6 that rough correspondence between the wave-tank and open ocean data in Fig.11 shows the lab results are capturing the essential mechanisms, since the mechanistic details in each case could be significantly different (the physical details certainly are) and the rough agreement coincidental. As someone with a keen interest in this topic but limited experience with of wind-wave tank experiments I'd like to see a more thorough examination of these issues.

First a comment to the significance of our lab measurements for open ocean conditions. We have done the first systematic study at all in the wind speed range beyond 33 m/s $u_{10}$. So far only three data points

with huge error bars were available as shown in Figure 11 of our paper. In the wind speed range, we found a very steep increase of the gas transfer velocities even without the effect of bubbles, being associated to various rapid surface fragmentation processes at the free surface. We do not claim that this effect happens in the very same way at the open ocean, but it will happen also there, indicating that also the transfer of all water-side controlled gases will be enhanced significantly. This is an important new finding in our view for the global fluxes between ocean and atmosphere.

It is evident that gas transfer velocity - wind speed relations cannot be transferred from a wind-wave flume to the ocean. This is just as wrong as using empirical gas transfer - wind speed relations from a collection of field experiments. However, we insist that laboratory measurements are invaluable to identify the mechanisms of air-sea gas transfer. Laboratory measurements are generally much more precise and accurate than any current field measuring techniques. It is possible to use much more tracers simultaneously. And it is easy to perform systematic studies. It is not required to perform perfect simulations. This will not be possible. It is just necessary to identify and quantify mechanisms, which can then be adapted to open ocean conditions.

There were two serious limitations in the past: The limited wind speeds and only low-fetch conditions. The first limitation is already gone with the Kyoto High Windspeed Facility and the Miami SUSTAIN Facility. The second one can be overcome in annular facilities such as the Heidelberg Air-Sea Interaction Facility, the Aeolotron (Fig. 3).

[Figure]

Figure 3: Heidelberg Aeolotron: Due to the infinite fetch of the 10 m diameter facility, long and steep breaking wind waves can be generated, much larger than in any linear facility.

We have already modified the Heidelberg Aeolotron to perform experiments at higher wind speeds. With a number of new experimental techniques, which we have started to test this year, we are currently preparing experiments to cover an unprecedented range of wave ages in laboratory experiments and thus hope that we can make a useful contribution to solve the wave age dependency of air-sea gas exchange.

**References**

Blomquist, B. W., Brumer, S. E., Fairall, C. W., Huebert, B. J., Zappa, C. J., Brooks, I. M., Yang, M., Bariteau, L., Prytherch, J., Hare, J. E., Czerski, H., Matei, A., and Pascal, R. W. (2017). Wind speed and sea state dependencies of air-sea gas transfer: results from the high wind speed gas exchange study (hiwings). *J. Geophys. Res.*, 122:8034–8062.

Deike, L., Lenain, L., and Melville, W. K. (2017). Air entrainment by breaking waves. *Geophys. Res. Lett.*, 44:3779–3787.

Donelan, M., Haus, B., Reul, N., Plant, W., Stiassnie, M., Graber, H., Brown, O., and Saltzman, E. (2004). On the limiting aerodynamic roughness of the ocean in very strong winds. *Geophys. Res. Lett.*, 31(18).

Garbe, C. S., Rutgersson, A., Boutin, J., Delille, B., Fairall, C. W., Gruber, N., Hare, J., Ho, D., Johnson, M., de Leeuw, G., Nightingale, P., Pettersson, H., Piskozub, J., Sahlee, E., Tsai, W., Ward, B., Woolf, D. K., and Zappa, C. (2014). Transfer across the air-sea interface. In Liss, P. S. and Johnson, M. T., editors, *Ocean-Atmosphere Interactions of Gases and Particles*, pages 55–112. Springer.

Iwano, K., Takagaki, N., Kurose, R., and Komori, S. (2013). Mass transfer velocity across the breaking air-water interface at extremely high wind speeds. *Tellus B*, 65:21341.

Iwano, K., Takagaki, N., Kurose, R., and Komori, S. (2014). Erratum: Mass transfer velocity across the breaking air-water interface at extremely high wind speeds. *Tellus B*, 66:25233.

Krall, K. E. and Jähne, B. (2014). First laboratory study of air-sea gas exchange at hurricane wind speeds. *Ocean Sci.*, 10:257–265.

Mischler, W. (2014). *Systematic Measurements of Bubble Induced Gas Exchange for Trace Gases with Low Solubilities*. Dissertation, Institut für Umweltphysik, Fakultät für Physik und Astronomie, Univ. Heidelberg.

Powell, M. D., Vickery, P. J., and Reinhold, T. A. (2003). Reduced drag coefficient for high wind speeds in tropical cyclones. *Nature*, 422:279–283.

Rhee, T., Nightingale, P., Woolf, D., Caulliez, G., Bowyer, P., and Andreae, M. (2007). Influence of energetic wind and waves on gas transfer in a large wind-wave tunnel facility. *J. Geophys. Res.*, 112:5027.

Takagaki, N., Komori, S., Suzuki, N., Iwano, K., Kuramoto, T., Shimada, S., Kurose, R., and Takahashi, K. (2012). Strong correlation between the drag coefficient and the shape of the wind sea spectrum over a broad range of wind speeds. *Geophys. Res. Lett.*, 39.

Zavarsky, A. (2018). *Eddy covariance air-sea gas flux measurements. Regional sources and gas transfer limitation*. PhD thesis.

Zavarsky, A., Goddijn-Murphy, L., Steinhoff, T., and Marandino, C. A. (2018). Bubble-mediated gas transfer and gas transfer suppression of dms and co2. *Journal of Geophysical Research: Atmospheres*, 123(12):6624–6647.

**Appendix**

The following plots will also appear in a supplement to the final revised paper.

[Figure]

Figure 4: Kyoto freshwater experiment: Modeled vs. measured transfer velocities, colors corresponding to the tracers (a) and colors corresponding to the wind speeds used (b). The solid line marks perfect agreement, the dashed lines plus or minus 15%. He was excluded from the fit, therefore it is only shown here with open symbols. (c) bubble surface transfer velocity $k_c$ in dependency of the solubility for the wind speeds, for which a bubble contribution was detected. The highest wind speed condition was repeated twice, one of the repetitions is shown as a dashed line, the other as a solid line.

[Figure]

Figure 5: Kyoto seawater model experiment: Modeled vs. measured transfer velocities, colors corresponding to the tracers (a) and colors corresponding to the wind speeds used (b). The solid line marks perfect agreement, the dashed lines plus or minus 15%. He was excluded from the fit, therefore it is only shown here with open symbols. (c) bubble surface transfer velocity $k_c$ in dependency of the solubility for the wind speeds, for which a bubble contribution was detected.

[Figure]

Figure 6: Miami seawater experiment: Modeled vs. measured transfer velocities, colors corresponding to the tracers (a) and colors corresponding to the wind speeds used (b). The solid line marks perfect agreement, the dashed lines plus or minus 15%. He was excluded from the fit, therefore it is only shown here with open symbols. (c) bubble surface transfer velocity $k_c$ in dependency of the solubility for the wind speeds, for which a bubble contribution was detected. The highest wind speed condition was repeated twice, one of the repetitions is shown as a dashed line, the other as a solid line.

---

## Author Comment (AC2) · 3 Sep 2019

**by Christopher Fairall**

September 3, 2019

The authors thank Christopher Fairall for his thorough review and helpful comments. A point by point answer to his questions and comments can be found below.

**Missing bubble contribution**

> Reviewer's comment: Their main finding – that bubble transfer is negligible compared to the direct transfer for all gases up to wind speeds of 30 m/s – is surprising (at least to this reviewer).

This was very unexpected for the authors as well. This is why we reported the Kyoto results only on conferences but did not write a paper and waited for the results of a second experiment in the much larger SUSTAIN facility with real seawater, where very similar results were obtained.

**Reconciliation of lab and field measurements**

> Reviewer's comment: The authors' conclusions are difficult to reconcile with open ocean measurements but I leave it to them to consider how to handle that. In my opinion they are too dismissive of the large number of experimental (tracers and eddy covariance) studies that indicate insoluble gases and CO2 have substantially higher transfer velocity above 15 m/s. I question that the Zavarsky paper is sufficient cover.

In the conclusions of our paper, we will put more emphasis on the unsolved problem of wave age or fetch dependency of the gas transfer velocity.

By the way, the authors are surprised that both reviewers do not comment at all our findings for wind speeds beyond 33 m/s, were we found a very steep increase of the gas transfer velocities even without the effect of bubbles, being associated with various rapid surface fragmentation processes at the free surface. We do not claim that this effect happens in the very same way at the open ocean, but it will happen also there, indicating that also the transfer of all water-side controlled gases will be enhanced significantly independently of solubility. In our view, this is an important new finding with relevance for the global fluxes between ocean and atmosphere.

**Computation of $u_{10}$ and momentum balance**

> Reviewer's comment: I don't understand why they used Powell's open ocean estimates of Cd when they could have used Donelan's 2004 results actually determined in the Miami wind tunnel. Also, please give us a sentence explaining Takagadi's method for getting friction velocity so we don't have to go look it up. I am guessing they assumed a momentum balance at the interface to compute u*w from u*a (square root of ratio of air to water density). This assumes that the growth of the wave field has negligible effect of the momentum balance. Is this right?

Also, they switch back and forth between u*w and just u* - I assume they mean the same thing. Just be advised that it is quite a stretch from a setting on an instrument dial to actual waterside friction velocity.
*and later* . . .
U*a is a measure for momentum extracted from the wind. Some of that momentum is realized in the ocean via direct viscous transfer at the interface, some goes into growing waves and some is later realized as turbulence when waves break. If locally waves are in dynamic balance, then the momentum flux from the air is the same as the momentum flux realized in the ocean. So, how close is the balance in a wind tunnel?

- We now explain this in more detail in the manuscript. To this end we will add our co-workers from Kyoto (Naohisa Takagaki) and Miami (Andrew Smith) as co-authors, because they made the measurements and greatly helped with computing $u_{10}$. We decided to use Donelan et al. (2004) (lab measurements of $C_D$) instead of Powell et al. (2003) (field measurements of $C_D$) to convert measured wind speeds in Miami to $u_{*,a}$ and $u_{10}$ and we also found a mistake in converting between $u_{*,a}$ and $u_{*,w}$, which we corrected for the final paper. This slightly changes the relationships $k_x(u_{*,w})$ and the model parameterization equations, but not our findings.

- In short fetch wind-wave tunnels, the growth of waves has a considerable effect on the momentum balance. At the moment, we have only data from one fetch, measured in the large Marseille facility and only being published in a PhD thesis (Bopp, 2018) with results similar to Banner and Peirson (1998). So the best we could do is to assume momentum balance. We wanted to use the water side friction velocity, because gas transfer is controlled there.

- Indeed, the label in Fig. 5 should read as $u_{*w}$ and not $u_*$ and will be changed in the final paper.

- As a side note, the wind-wave tank that Donelan et al. (2004) used was a different one than the one used in this study (SUSTAIN). So Donelan et al. (2004) did not determine $C_d$ in **the** Miami wind wave tunnel, but merely in **a** Miami wind wave tunnel.

- In the fetch-limited SUSTAIN tank, waves are growing until their structure changes/collapses due to momentum transfer (forcing) from the wind or breaking and they do not reach true equilibrium where air-side and water-side stress are exactly equal. An upcoming experiment in SUSTAIN on the drag coefficient and momentum balance (lab vs. field) is presently in planning stages.

**DMS and carbon dioxide**

Reviewer's comment: Another example, McNeil and D'Asaro (2007) did not 'measure' gas transfer velocities, they inferred them from water concentration measurements. Furthermore, their basic assumption in the analysis is that both free surface and bubble transfer velocities scale with u*w and nothing else. However, there is considerable evidence that the air volume flux from breaking scales as u* and other wave parameters (see Deike et al. 2017). Also note that Deike and Melville (2018) used this approach to estimate kb for DMS and CO2 – treating both gases as highly soluble. They present measurements of bubble area from the wind tunnels but no estimates of volume flux. I think the bubble volume flux data should appear in Fig. 7c. Finally, the authors might note that Rhee et al. 2007 found considerable enhancement of k for insoluble gases when bubbles were introduced.
*and later* . . .
The discussion of field measurements of k for DMS and CO2 is very useful (Section 4.5). It also illustrates the rather inconclusive state of field observations. The data from Zavarsky et al (2018) show essentially no difference between CO2 and DMS. The analysis of Fairall et al. (2011) which compiled all the direct flux data to date showed significant differences. The HIWINGS data shown in this figure are quite surprising for CO2. Blomqvist gives k CO2 a power dependence of u10^1.68, which is not linear. Because of the conditions, I don't think the HIWINGS data below U10 of 10 m/s should be considered. Even ancient information such

as Wanninkhoff's famous formula indicate a quadratic wind speed dependence for insoluble gases. Earlier suggestions that k CO2 should scale as U^3 were based on the assumption that whitecap fraction scaled as U^3. More recent observations have shown that this is not the case, with much weaker wind speed dependence at high wind speeds. I think the authors may be placing too much importance on Zavarsky. From Wanninkhoff's radioactive tracers, to a number of deliberate tracer studies, and perhaps all other eddy flux measurements CO2 goes at least quadratic with wind speed. At u10=18 m/s, the value is close to 100 cm/h for open ocean measurements.

- For the data of McNeil and D'Asaro (2007), we will change 'measured' to 'estimated'.

- We are surprised about the statement that the bubble volume flux $V_a$ should scale linearly with $u_*$. Did you really mean this? The equations in Fig. 4 in Deike et al. (2017) rather says $V_a = 2.3 \cdot 10^{-3} c_p^{-0.9} u_*^{1.9}$. By the way, this finding is just the opposite of the finding by Blomquist et al. (2017). They found lower gas transfer velocities in 'young' sea states than in 'old' seas for carbon dioxide and attributed this to higher bubble contributions at older seas. Deike et al. (2017) found that the air entrainment is much lower at high wave ages. The effect is large, air entrainment scales roughly with the inverse wave age. Thus the wave age dependency is still an open question and further investigations are required.

- Rhee et al. (2007) is, among other studies, rather irrelevant for our work, because 1) their highest measured wind speed is 13 m/s, and 2) their means of bubble generation (submerged aerators) is very different from ours (wave breaking induced bubbles only). Therefore such a comparison is not meaningful. In the introduction, we refer, of course, to the two previous lab studies in the Kyoto high wind speed facility: Iwano et al. (2013, 2014) and Krall and Jähne (2014).

- We selected the data of Blomquist et al. (2017) because it had the highest wind speeds up to 25 m/s and those of Zavarsky et al. (2018) because they show no difference is the gas exchange rates between DMS and carbon dioxide. We place equal importance to both field studies, and can only state that there are contradicting results. Unfortunately, eddy covariance measurements are still not precise enough (especially compared to gas transfer velocity measurements in the lab) and prone to systematic errors. Our current conclusion is therefore, that there are significant yet unresolved wave age and sea state effects. We will emphasize this more clearly in the conclusions and the abstract of our paper.

- We will add a short paragraph to the manuscript, discussing transfer velocities measured using the dual tracer technique with He and $SF_6$, which have a very low solubility and should, therefore, have a larger bubble effect than $CO_2$. However, $SF_6$ and He were found to have lower transfer velocities than $CO_2$ measured with eddy covariance, see the compilation of field measurements in Garbe et al. (2014, Fig. 2.10a). This is another contradicting field result.

**Minor comments**

Reviewer's comment: Bubbles may also suppress turbulence through density stratification.

Correct, but for the wind-wave tanks with just below one meter depth, this effect is not significant at all for our measurements.

Reviewer's comment: Eq (5) This terminology is confusing with the un-numbered equation on Page 2 line 14.

We changed the wording in the first paragraph of Sec. 2.1 for clarity.

Reviewer's comment: Page 4 line 1 Suggest referencing bubble model work of Liang et al. GLOBAL BIO- GEOCHEMICAL CYCLES, VOL. 27, 894–905, doi:10.1002/gbc.20080, 2013 and ear- lier work.

We decided not to cite Liang et al. (2013), because their focus is more on supersaturation and fluxes and they did not measure the transfer velocity.

> Reviewer's comment: Page 5 Eq (7) I am confused by the terminology. Can Qb be volume flux and Qb/As also be a volume flux? If we equate kr with the volume of air ingested per unit area per unit time (units velocity), then that should be on the order of 30 cm/hr at u10=15 m/s and u*w=2 cm/s (see, Deike et al, 2017). That does not compare well with Fig. 7 c, where kr doesn't reach those values until u*w is greater than 10.

- $Q_b/A_s$ has the units of a velocity. We will change its name in accordance with Deike et al. (2017) to 'air entrainment velocity'.

- Deike et al. (2017) found a considerable wave age effect and our measurements add another value for very low wave ages not covered by the data Deike is using.

> Reviewer's comment: Page 5 Eq (10) This equation is similar to Woolf parameterization. For the volume flux is kr=2450*whitecap fraction (cm/h) and the parameter e=kc/kr*sqrt(600). How does this compare to your results?

We can't directly compare because we have not yet estimated the whitecap fraction. We will do this in a further paper once we have analyzed the images from the water surface. Currently, we can only say that the solubility dependency is about the same. However, Mischler (2014) has shown in a bubble tank study, that the Woolf et al. (2007) and the Mischler (2014) parameterizations for the bubble mediated transfer velocity $k_c$ perform equally well, however, the Mischler (2014) parameterization uses one fewer parameter.

> Reviewer's comment: Page 7 section 2.3. This discussion of droplet effects is a little confusing. It seems to me that ejecting a droplet does not change the waterside concentration, so their measurement method does not capture it. If the drop has time, it would transfer gas to the air and that would reduce free surface transfer further down the line. Is that what they are trying to say? This argument about time scales ignores the fact that the droplets leave the wind tunnel before they can do much transferring – this is discussed in Andreas and Mahrt 2016.

Even though the fetch of the wind-wave tanks is rather short compared to open ocean conditions, some of the droplets generated, will impact the water surface again. Andreas et al. (2017) discusses the time scales involved in great detail. They state, that only for the largest radii and for weak winds less than 15 m/s, the droplets fall back into the ocean before they establish equilibrium with the atmospheric gas reservoir. So even though not all spray droplets reimpact the water surface since they are blown out of the wind wave tank, those that do impact the water again do change the water side concentration, since they will have equilibrated with the air.

Since, according to Andreas et al. (2017), at extreme conditions, the spray droplets do come into equilibrium before falling back into the water, the diffusion coefficient of the tracer no longer plays a role in spray mediated gas transfer. Therefore, Helium, despite having a much higher diffusion coefficient than other gases, will no longer have a correspondingly faster transfer velocity across the pray droplet interface, since this transfer velocity only depends on how much spray is generated.

> Reviewer's comment: Figure 10. What drag coefficient is used for the curve shown for modeled DMS and CO2?

We used the $u_{10}$-$u_{*,w}$ relationship shown as a gray line in Fig. 4a, which corresponds to the drag coefficient shown as the gray line in Fig. 1 below.

[Figure]

Figure 1: The gray line shows the drag coefficient used for converting $u_{*,w}$ to $u_{10}$ for the Figs. 9, 10 and 11 in the manuscript.

**References**

Andreas, E. L., Vlahos, P., and Monahan, E. C. (2017). Spray-mediated air-sea gas exchange: the governing time scales. *J. Mar. Sci. Eng.*, 5:60.

Banner, M. L. and Peirson, W. L. (1998). Tangential stress beneath wind-driven air-water interfaces. *J. Fluid Mech.*, 364:115–145.

Blomquist, B. W., Brumer, S. E., Fairall, C. W., Huebert, B. J., Zappa, C. J., Brooks, I. M., Yang, M., Bariteau, L., Prytherch, J., Hare, J. E., Czerski, H., Matei, A., and Pascal, R. W. (2017). Wind speed and sea state dependencies of air-sea gas transfer: results from the high wind speed gas exchange study (hiwings). *J. Geophys. Res.*, 122:8034–8062.

Bopp, M. (2018). *Air-Flow and Stress Partitioning over Wind Waves in a Linear Wind-Wave Facility*. Dissertation, Institut für Umweltphysik, Fakultät für Physik und Astronomie, Univ. Heidelberg, Heidelberg.

Deike, L., Lenain, L., and Melville, W. K. (2017). Air entrainment by breaking waves. *Geophys. Res. Lett.*, 44:3779–3787.

Donelan, M., Haus, B., Reul, N., Plant, W., Stiassnie, M., Graber, H., Brown, O., and Saltzman, E. (2004). On the limiting aerodynamic roughness of the ocean in very strong winds. *Geophys. Res. Lett.*, 31(18).

Garbe, C. S., Rutgersson, A., Boutin, J., Delille, B., Fairall, C. W., Gruber, N., Hare, J., Ho, D., Johnson, M., de Leeuw, G., Nightingale, P., Pettersson, H., Piskozub, J., Sahlee, E., Tsai, W., Ward, B., Woolf, D. K., and Zappa, C. (2014). Transfer across the air-sea interface. In Liss, P. S. and Johnson, M. T., editors, *Ocean-Atmosphere Interactions of Gases and Particles*, pages 55–112. Springer.

Iwano, K., Takagaki, N., Kurose, R., and Komori, S. (2013). Mass transfer velocity across the breaking air-water interface at extremely high wind speeds. *Tellus B*, 65:21341.

Iwano, K., Takagaki, N., Kurose, R., and Komori, S. (2014). Erratum: Mass transfer velocity across the breaking air-water interface at extremely high wind speeds. *Tellus B*, 66:25233.

Krall, K. E. and Jähne, B. (2014). First laboratory study of air-sea gas exchange at hurricane wind speeds. *Ocean Sci.*, 10:257–265.

Liang, J.-H., Deutsch, C., McWilliams, J. C., Baschek, B., Sullivan, P. P., and Chiba, D. (2013). Parameterizing bubble-mediated air-sea gas exchange and its effect on ocean ventilation. *Global Biogeochem. Cycles*, 27(3):894–905.

McNeil, C. and D'Asaro, E. (2007). Parameterization of air sea gas fluxes at extreme wind speeds. *J. Marine Syst.*, 66:110–121.

Mischler, W. (2014). *Systematic Measurements of Bubble Induced Gas Exchange for Trace Gases with Low Solubilities.* Dissertation, Institut für Umweltphysik, Fakultät für Physik und Astronomie, Univ. Heidelberg.

Powell, M. D., Vickery, P. J., and Reinhold, T. A. (2003). Reduced drag coefficient for high wind speeds in tropical cyclones. *Nature*, 422:279–283.

Rhee, T., Nightingale, P., Woolf, D., Caulliez, G., Bowyer, P., and Andreae, M. (2007). Influence of energetic wind and waves on gas transfer in a large wind-wave tunnel facility. *J. Geophys. Res.*, 112:5027.

Woolf, D., Leifer, I., Nightingale, P., Rhee, T., Bowyer, P., Caulliez, G., de Leeuw, G., Larsen, S., Liddicoat, M., Baker, J., and Andreae, M. (2007). Modelling of bubble-mediated gas transfer: Fundamental principles and a laboratory test. *J. Marine Syst.*, 66:71–91.

Zavarsky, A., Goddijn-Murphy, L., Steinhoff, T., and Marandino, C. A. (2018). Bubble-mediated gas transfer and gas transfer suppression of dms and co2. *Journal of Geophysical Research: Atmospheres*, 123(12):6624–6647.